# AMAGO: Scalable In-Context Reinforcement Learning for Adaptive Agents

**Jake Grigsby**[1], **Linxi "Jim" Fan**[2], **Yuke Zhu**[1]
[1]The University of Texas at Austin    [2]NVIDIA Research

## Abstract

We introduce AMAGO, an in-context Reinforcement Learning (RL) agent that uses sequence models to tackle the challenges of generalization, long-term memory, and meta-learning. Recent works have shown that off-policy learning can make in-context RL with recurrent policies viable. Nonetheless, these approaches require extensive tuning and limit scalability by creating key bottlenecks in agents' memory capacity, planning horizon, and model size. AMAGO revisits and redesigns the off-policy in-context approach to successfully train long-sequence Transformers over entire rollouts in parallel with end-to-end RL. Our agent is scalable and applicable to a wide range of problems, and we demonstrate its strong performance empirically in meta-RL and long-term memory domains. AMAGO's focus on sparse rewards and off-policy data also allows in-context learning to extend to goal-conditioned problems with challenging exploration. When combined with a multi-goal hindsight relabeling scheme, AMAGO can solve a previously difficult category of open-world domains, where agents complete many possible instructions in procedurally generated environments.

## 1 Introduction

Reinforcement Learning (RL) research has created effective methods for training *specialist* decision-making agents that maximize one objective in a single environment through trial-and-error [1, 2, 3, 4, 5]. New efforts are shifting towards developing *generalist* agents that can adapt to a variety of environments that require long-term memory and reasoning under uncertainty [6, 7, 8, 9]. One of the most promising approaches to generalist RL is a family of techniques that leverage sequence models to actively adapt to new situations by learning from experience at test-time. These "in-context" agents use memory to recall information from previous timesteps and update their understanding of the current environment [10, 11]. In-context RL's advantage is its simplicity: it reduces partial observability, generalization, and meta-learning to a single problem in which agents equipped with memory train in a collection of related environments [12, 13, 14]. The ability to explore, identify, and adapt to new environments arises implicitly as agents learn to maximize returns in deployments that may span multiple trials [13, 15, 16].

Effective in-context agents need to be able to scale across two axes: 1) the length of their planning horizon, and 2) the length of their effective memory. The earliest versions of this framework [10, 11, 17] use recurrent networks and on-policy learning updates that do not scale well across either axis. Transformers [18] improve memory by removing the need to selectively write information many timesteps in advance and turning recall into a retrieval problem [19]. Efforts to replace recurrent policies with Transformers were successful [20, 21], but long-term adaptation remained challenging, and the in-context approach was relegated to an unstable baseline for other meta-RL methods [12]. However, it has been shown that pure in-context RL with recurrent networks can be a competitive baseline when the original on-policy gradient updates are replaced by techniques from a long line of work in stable off-policy RL [22, 23]. A clear next step is to reintroduce the memory benefits of Transformers. Unfortunately, training Transformers over long sequences with off-policy RL combines two of the most implementation-dependent areas in the field, and many of the established best practices do not scale (Section 3). The difficulty of this challenge is perhaps best highlighted by the fact that many popular applications of Transformers in off-policy (or offline) RL devise ways to avoid RL altogether and reformulate the problem as supervised learning [24, 25, 26, 27, 28, 29].

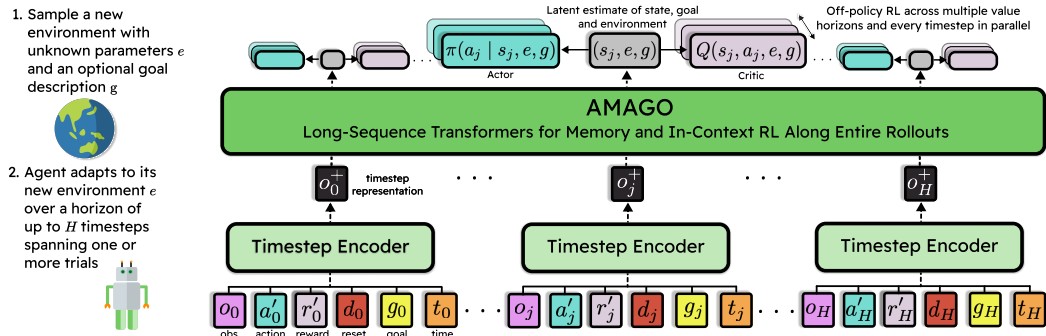

Figure 1: In-context RL techniques solve memory and meta-learning problems by using sequence models to infer the identity of unknown environments from test-time experience. AMAGO addresses core technical challenges to unify the performance of end-to-end off-policy RL with long-sequence Transformers in order to push memory and adaptation to new limits.

Our work introduces AMAGO (Adaptive Memory Agent for achieving GOals) — a new learning algorithm that makes two important contributions to in-context RL. First, we redesign the off-policy actor-critic update from scratch to support long-sequence Transformers that learn from entire rollouts in parallel (Figure 1). AMAGO breaks off-policy in-context RL's biggest bottlenecks by enabling memory lengths, model sizes, and planning horizons that were previously impractical or unstable. Our agent is open-source[1] and specifically designed to be efficient, stable, and applicable to new environments with little tuning. We empirically demonstrate its power and flexibility in existing meta-RL and memory benchmarks, including state-of-the-art results in the POPGym suite [30], where Transformers' recall dramatically improves performance. Our second contribution takes in-context RL in a new direction with fresh benchmarks. AMAGO's use of off-policy data — as well as its focus on long horizons and sparse rewards — makes it uniquely capable of extending the in-context RL framework to goal-conditioned problems [31, 32] with hard exploration. We add a new hindsight relabeling scheme for *multi-step* goals that generates effective exploration plans in multi-task domains. AMAGO can then learn to complete many possible instructions while using its memory to adapt to unfamiliar environments. We first study AMAGO in two new benchmarks in this area before applying it to instruction-following tasks in the procedurally generated worlds of Crafter [33].

## 2 RELATED WORK

**RL Generalization and Meta-Learning.** General RL agents should be able to make decisions across similar environments with different layouts, physics, and visuals [34, 35, 36, 37]. Many techniques are based on learning invariant policies that behave similarly despite these changes [6]. Meta-RL takes a more active approach and studies methods for rapid adaptation to new RL problems [38]. Meta-RL is a crowded field with a complex taxonomy [12]; this paper uses the informal term "in-context RL" to refer to a specific subset of *implicit context-based* methods that treat meta-learning, zero-shot generalization, and partial observability as a single problem. In practice, these methods are variants of RL$^2$ [11, 10], where we train a sequence model with standard RL and let memory and meta-reasoning arise implicitly as a byproduct of maximizing returns [16, 21, 17, 22, 15, 23, 8]. There are many other ways to use historical context to adapt to an environment. However, in-context RL makes so few assumptions that alternatives mainly serve as ways to take advantage of problems that do not need its flexibility. Examples include domains where each environment is fully observed, can be labeled with ground-truth parameters indicating its unique characteristics, or does not require meta-adaptation within trials [39, 40, 41, 42, 43, 44].

The core AMAGO agent is conceptually similar to AdA [8] — a recent work on off-policy meta-RL with Transformers. AdA trains in-context RL agents in a closed-source domain that evaluates task diversity at scale, while our work is more focused on long-term recall and open-source RL engineering details.

---

[1]Code is available here: https://ut-austin-rpl.github.io/amago/

**Goal-Conditioned RL.** Goal-conditioned RL (GCRL) [31, 32] trains multi-task agents that generalize across tasks by making the current goal an input to the policy. GCRL often learns from sparse (binary) rewards based solely on task completion, reducing the need for reward engineering. GCRL methods commonly use hindsight experience replay (HER) [45, 46, 47, 48] to ease the challenges of sparse-reward exploration. HER randomly relabels low-quality data with alternative goals that lead to higher rewards, and this technique is mainly compatible with off-policy methods where the same trajectory can be recycled many times with different goals. Goal conditioned supervised learning (GCSL) [49, 50, 51, 52, 53, 54] uses HER to relabel every trajectory and directly imitates behavior conditioned on future outcomes. GCSL has proven to be an effective way to train Transformers for decision-making [26, 24, 27, 25, 55, 56] but has a growing number of theoretical and practical issues [57, 58, 59, 60]. AMAGO avoids these limitations by providing a stable and effective way to train long-sequence Transformers with pure RL.

## 3 BACKGROUND AND PROBLEM FORMULATION

**Solving Goal-Conditioned CMDPs with In-Context RL.** Partially observed Markov decision processes (POMDPs) create long-term memory problems where the true environment state $s_t$ may need to be inferred from a history of limited observations $(o_0, \ldots, o_t)$ [61]. Contextual MDPs (CMDPs) [62, 6, 63] extend POMDPs to create a distribution of related decision-making problems where everything that makes an environment unique can be identified by a *context parameter*. More formally, an environment's context parameter, $e$, conditions its reward, transition, observation emission function, and initial state distribution. CMDPs create zero-shot generalization problems when $e$ is sampled at the end of every episode [6] and meta-RL problems when held constant for multiple trials [12]. We slightly adapt the usual CMDP definition by decoupling an optional *goal* parameter $g$ from the environment parameter $e$ [41]. The goal $g$ only contributes to the reward function $R_{e,g}(s, a)$, and lets us provide a task description without assuming we know anything else about the environment.

Knowledge of $e$ lets a policy adapt to any given CMDP, but its value is not observed as part of the state (and may be impossible to represent), creating an "implicit POMDP" [14, 39]. From this perspective, the only meaningful difference between POMDPs and CMDPs is the need to reason about transition dynamics $(o_i, a_i, o_{i+1})$, rewards $(r)$, and resets $(d)$ based on full trajectories of environment interaction $\tau_{0:t} = (o_0, a_0, r_1, d_1, o_1, a_1, r_2, d_2, \ldots, o_t)$. In-context RL agents embrace this similarity by treating CMDPs as POMDPs with additional inputs and using their memory to update state and environment estimates at every timestep. Memory-equipped policies $\pi$ take trajectory sequences as input and output actions that maximize returns over a finite horizon $(H)$ that may span multiple trials:
$$\pi^* = \text{argmax}_\pi \, \mathbb{E}_{p(e,g), a_t \sim \pi(\cdot|\tau_{0:t}, g)} \left[ \sum_{t=0}^{H} R_{e,g}(s_t, a_t) \right].$$

**Challenges in Off-Policy Model-Free In-Context RL.** In-context RL is a simple and powerful idea but is often outperformed by more complex methods [64, 42, 12]. However, deep RL is a field where implementation details are critical [65], and recent work has highlighted how the right combination of off-policy techniques can close this performance gap [22]. Off-policy in-context agents collect training data in a replay buffer from which they sample trajectory sequences $\tau_{t-l:t} = (o_{t-l}, a_{t-l}, r_{t-l+1}, \ldots, o_t)$ up to a fixed maximum *context length* of $l$. Training on shorter sequences than used at test-time is possible but creates out-of-distribution behavior [66, 67, 68]. Therefore, an agent's context length is its most important bottleneck because it establishes an upper bound on memory and in-context adaptation. Model-free RL agents control their planning horizon with a value learning discount factor $\gamma \in [0, 1)$. Despite context-based learning's focus on long-term adaptation, learning instability restricts nearly all methods to $\gamma \le .99$ and effective planning horizons of around 100 timesteps. Trade-offs between context length, planning horizon, and stability create an unpredictable hyperparameter landscape [22] that requires extensive tuning.

Agents that are compatible with both discrete and continuous actions use $\tau_{t-l:t}$ to update actor $(\pi)$ and critic $(Q)$ networks with two loss functions optimized in alternating steps [69, 70] that make it difficult to share sequence model parameters [22, 71, 72]. It is also standard to use multiple critic networks to reduce overestimation [73, 74] and maintain extra target networks to stabilize optimization [3]. The end result is several sequence model forward/backward passes per training step — limiting model size and context length [75, 22] (Figure 10). This problem is compounded by

the use of recurrent networks that process each timestep sequentially [76]. Transformers parallelize sequence computation and improve recall but can be unstable in general [77, 78], and the differences between supervised learning and RL create additional challenges. RL training objectives do not always follow clear scaling laws, and the correct network size can be unpredictable. Datasets also grow and shift over time as determined by update-to-data ratios that impact performance but are difficult to tune [79, 80]. Finally, the rate of policy improvement changes unpredictably and makes it difficult to prevent model collapse with learning rate cooldowns [18, 81]. Taken together, these challenges make us likely to optimize a large policy network for far longer than our small and shifting dataset can support [82, 83]. Addressing these problems requires careful architectural changes [20, 84] and Transformers are known to be an inconsistent baseline in our setting [21, 28, 8].

## 4 METHOD

We aim to extend the limits of off-policy in-context RL's three main barriers: 1) the memory limit or context length $l$, 2) the value discount factor $\gamma$, and 3) the size and recall of our sequence model. Transformers are a strong solution to the last challenge and may be able to address all three barriers if we were able to learn from long context lengths $l \approx H$ and select actions for $\gamma \geq .999$ at test time. In principle, this would allow agents to remember and plan for long adaptation windows in order to scale to more challenging problems while removing the need to tune trade-offs between stability and context length. AMAGO overcomes several challenges to make this possible. This section summarizes the most important techniques that enable AMAGO's performance and flexibility. More details can be found in Appendix A and in our open-source code release.

A high-level overview of AMAGO is illustrated in Figure 1. The observations of every environment are augmented to a unified goal-conditioned CMDP format, even when some of the extra input information is unnecessary. This unified format allows AMAGO to be equally applicable to generalization, meta-RL, long-term memory, and multi-task problems without changes. AMAGO is an off-policy method that takes advantage of large and diverse datasets; trajectory data is loaded from disk and can be relabeled with alternative goals and rewards. The shape and format of trajectories vary across domains, but we use a timestep encoder network to map each timestep to a fixed-size representation. AMAGO allows the timestep encoder to be the only architecture change across experiments. A single Transformer trajectory encoder processes sequences of timestep representations. AMAGO uses these representations as the inputs to small feed-forward actor and critic networks, which are optimized simultaneously across every timestep of the causal sequence. Like all implicit context-based methods (Sec. 2), this architecture encourages the trajectory encoder outputs to be latent estimates of the current state and environment $(s_i, e)$ (Sec. 3). In short, AMAGO's learning update looks more like supervised sequence modeling than an off-policy actor-critic, where each training step involves exactly one forward pass of one Transformer model with two output heads. This end result is simple and scalable but is only made possible by important technical details.

**Sharing One Sequence Model.**   The accepted best practice in off-policy RL is to optimize separate sequence models for the actor and critic(s) — using additional target models to compute the critic loss. Sharing actor and critic sequence models is possible but has repeatedly been shown to be unstable [22, 71, 72]. AMAGO addresses this instability and restructures the learning update to train its actor and critics on top of the same sequence representation; there are no target sequence models, and every parameter is updated simultaneously without alternating steps or conflicting learning rates. We combine the actor and critic objectives into a single loss trained by one optimizer. However, we must put each term on a predictable scale to prevent their weights from needing to be tuned across every experiment [85, 86]. The key detail enabling the simultaneous update to work in AMAGO — without separating the Transformer from the actor loss [72, 71] — is to carefully detach the critic from the actor loss (Appendix A.1). We compute loss terms with a custom variant of REDQ [74] for discrete and continuous actions that removes unintuitive hyperparameters wherever possible.

**Stable Long-Context Transformers in Off-Policy RL.**   Transformers bring additional optimization questions that can be more challenging in RL than supervised learning (Sec. 2). The shared off-policy update may amplify these issues, as we were unable to successfully apply existing architectural changes for Transformers in on-policy RL [84, 20] to our setting. We find that *attention entropy collapse* [87] is an important problem in long-sequence RL. Language modeling and other successful applications of Transformers involve learning a variety of temporal patterns. In contrast, RL

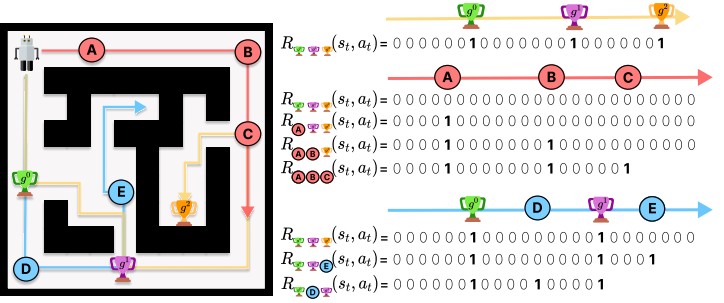

Figure 2: The agent (top left) navigates a maze to reach goal locations ($g^0, g^1, g^2$). Yellow, red, and blue paths with locations (A, ..., E) are examples of trajectories and achieved goals for relabeling. We show how instruction-relabeling creates a variety of alternative reward sequences.

agents can converge on precise memory strategies that consistently recall specific timesteps of long sequences (Figure 15). Optimizing these policies encourages large dot products between a small set of queries and keys that can destabilize attention[2]. We modify the Transformer architecture based on Normformer [88] and $\sigma$Reparam [87], and replace saturating activations with Leaky ReLUs to preserve plasticity [83, 89] as discussed in Appendix A.3. This architecture effectively stabilizes training and reduces tuning by letting us pick model sizes that are safely too large for the problem.

**Long Horizons and Multi-Gamma Learning.** While long-sequence Transformers improve recall, adaptation over extended rollouts creates a challenging credit assignment and planning problem. An important barrier in increasing adaption horizons is finding stable ways to increase the discount factor $\gamma$. Time information in the trajectory improves stability [90], but we also use a "multi-gamma" update that jointly optimizes many different values of $\gamma$ in parallel. Each $\gamma$ creates its own $Q$-value surface, making the shared Transformer more likely to have a strong actor-critic learning signal for some $\gamma$ throughout training. The relative cost of the multi-gamma update becomes low as the size of the sequence model increases. AMAGO can select the action corresponding to any $\gamma$ during rollouts, and we use $\gamma \geq .999$ unless otherwise noted. A discrete-action version of the multi-gamma update has previously been studied as a byproduct of hyperbolic discounting in (memory-free) MDPs with shared visual representations [91]. As a fail-safe, we include a filtered behavior cloning (BC) term [92, 93], which performs supervised learning when actions are estimated to have positive advantage ($Q(s, a) - V(s) > 0$) but does not directly depend on the scale of the $Q$ function.

**Hindsight Instruction Relabeling.** AMAGO's use of off-policy data and high discount factors allows us to relabel long trajectories in hindsight. This ability lets us expand the in-context RL framework to include goal-conditioned problems with sparse rewards that are too difficult for on-policy agents with short planning horizons to explore. AMAGO introduces a variant of HER (Sec. 3) for *multi-step* goals where $g = (g^0, \ldots, g^k)$. Besides adding the flexibility to create more complex multi-stage tasks, our experiments show that relabeling multi-step goals effectively creates automatic exploration plans in open-world domains like Crafter [33].

Our relabeling scheme works in two steps. First, we do not restrict goals to represent target states and allow them to be an abstract set of tokens or strings from a closed vocabulary. During rollouts, we can evaluate the goal tokens we were instructed to achieve and record alternatives that can be used for relabeling. The second step extends the HER relabeling scheme to "instructions", or sequences of up to $k$ subgoals. If an agent is tasked with $k$ goals but only completes the first $n \leq k$ steps of the instruction, we sample $h \in [0, k-n]$ timesteps that provide alternative goals, and then sample from all the goals achieved at those timesteps. We merge the $h$ alternative goals into the original instruction in chronological order and replay the trajectory from the beginning, recomputing the reward for the new instruction, which leads to a return of $n+h$ when rewards are binary. Figure 2 illustrates the technique with a maze-navigation example. Our implementation also considers importance-sampling variants that weight goal selection based on rarity and improve sample efficiency. Details are discussed in Appendix B.

---

[2]Environments that require little memory create a similar problem to those that demand long-term memory of specific information. Long context sequences create narrow attention distributions in both cases.

## 5 EXPERIMENTS

Our experiments are divided into two parts. First, we evaluate our agent in a variety of existing long-term memory, generalization, and meta-learning environments. We then explore the combination of AMAGO's adaptive memory and hindsight instruction relabeling in multi-task domains with procedurally generated environments. Additional results, details, and discussion for each of our experiments can be found in Appendix C. Unless otherwise noted, AMAGO uses a context length $l$ equal to the entire rollout horizon $H$. These memory lengths are not always necessary on current benchmarks, but we are interested in evaluating the performance of RL on long sequences so that AMAGO can serve as a strong baseline in developing new benchmarks that require more adaptation.

### 5.1 LONG-TERM MEMORY, GENERALIZATION, AND META-LEARNING

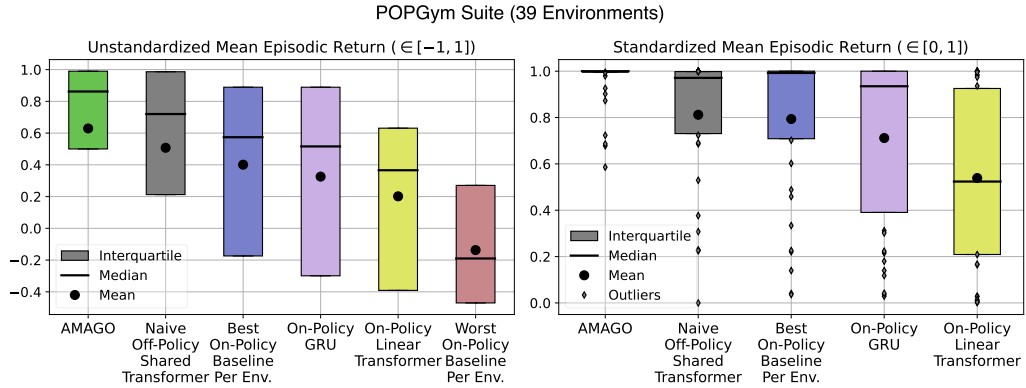

Figure 3: **Summary of POPGym Suite Results. (Left)** Aggregate results based on raw returns. **(Right)** Relative performance standardized by the highest and lowest scores in each environment.

POPGym [30] is a recent benchmark for evaluating long-term memory and generalization in a range of CMDPs. Performance in POPGym is far from saturated, and the challenges of sequence-based learning often prevent prior on-policy results from meaningfully outperforming memory-free policies. We tune AMAGO on one environment to meet the sample limit and architecture constraints of the original benchmark before copying these settings across 38 additional environments. Figure 3 summarizes results across the suite. We compare against the best existing baseline (a recurrent GRU-based agent [94]) and the most comparable architecture (an efficient Transformer variant [95]). We also report the aggregated results of the best baseline in each environment — equivalent to an exhaustive grid search over 13 alternative sequence models trained by on-policy RL. If we standardize the return in each environment to $[0, 1]$ based on the highest and lowest baseline, as done in [30], AMAGO achieves an average score of .95 across the suite, making it a remarkably strong default for sequence-based RL. We perform an ablation that naively applies the shared Transformer learning update without AMAGO's other details — such as our Transformer architecture and multi-gamma update. The naive baseline performs significantly worse, and the metrics in Figure 3 mask its collapse in 9/39 environments; AMAGO maintains stability in all of its 120+ trials despite using model sizes and update-to-data ratios that are not tuned. Learning curves and additional experiments are listed in Appendix C.1.

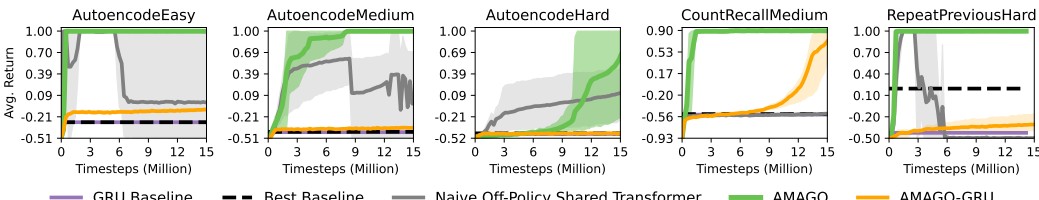

Figure 4: **Memory-Intensive POPGym Environments.** AMAGO's off-policy updates can improve performance in general, but its Transformer turns hard memory-intensive environments into a straight-forward recall exercise. Appendix C.1 reports results on more than 30 additional environments.

Much of AMAGO's performance gain in POPGym appears to be due to its ability to unlock Transformers' memory capabilities in off-policy RL: there are 9 recall-intensive environments where the GRU baseline achieves a normalized score of .19 compared to AMAGO's .999. Figure 4 ablates the impact of AMAGO's trajectory encoder from the rest of its off-policy details by replacing its Transformer with a GRU-based RNN in a sample of these recall-intensive environments.

We test the limits of AMAGO's recall using a "Passive T-Maze" environment from concurrent work [96], which creates a way to isolate memory capabilities at any sequence length. Solving this task requires accurate recall of the first timestep at the last timestep $H$. AMAGO trains a single RL Transformer to recover the optimal policy until we run out of GPU memory at context length $l = H = 10,000$ (Figure 5 top right). One concern is that AMAGO's maximum context lengths, large policies, and long horizons may hinder performance on easier problems where they are unnecessary, leading to the kind of hyperparameter tuning we would like to avoid. We demonstrate AMAGO on two continuous-action meta-RL problems from related work with dense (Fig. 5 left) and sparse (Fig. 5 bottom right) rewards where performance is already saturated, with strong and stable results.

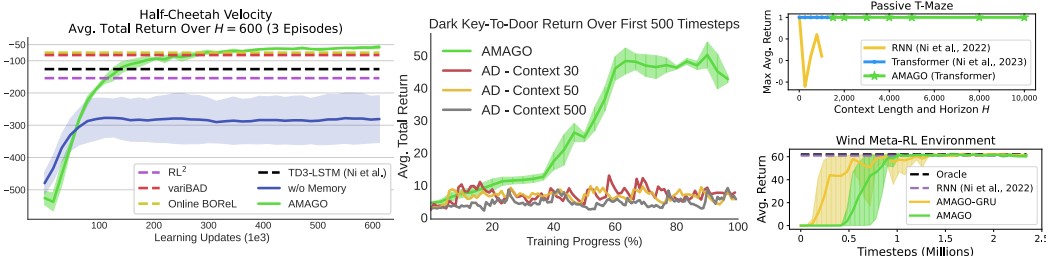

Figure 5: **Case Studies in In-Context Adaptation. (Left)** Adaptation over three episodes in Half-Cheetah Velocity [97]. **(Center)** AMAGO vs. Algorithm Distillation over the first 500 timesteps of Dark Key-To-Door [28]. **(Bottom Right)** Adaptation to a sparse-reward environment with continuous actions [22]. **(Top Right)** AMAGO solves the Passive T-Maze memory experiment [96] up until the GPU memory limit of $l = H = 10,000$ timesteps.

Like several recent approaches that reformulate RL as supervised learning (Sec. 2), AMAGO scales with the size and sequence length of a single Transformer. However, it creates a stable way to optimize its Transformer on the true RL objective rather than a sequence modeling loss. This can be an important advantage when the supervised objective becomes misaligned with our true goal. For example, we replicate Algorithm Distillation (AD) [28] on a version of its Dark Key-To-Door meta-learning task. AMAGO's ability to successfully optimize the return over any horizon $H$ lets us control sample efficiency at test-time (Fig. 5 center), while AD's adaptation rate is limited by its dataset and converges several thousand timesteps later in this case.

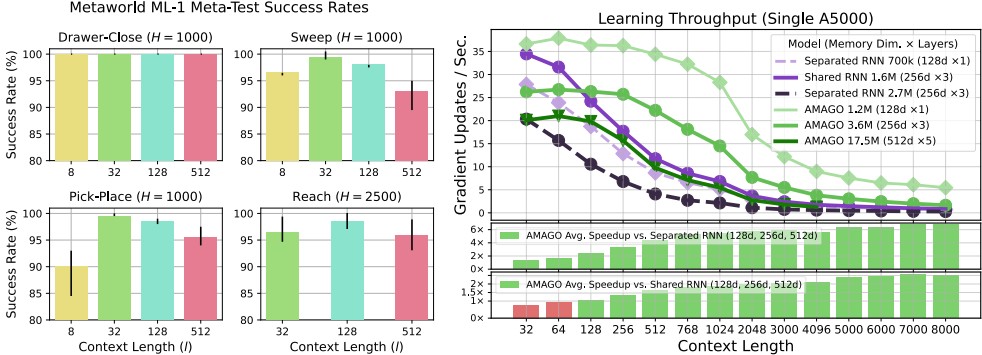

Figure 6: **(Left)** Meta-World [98] ML-1 success rate on held-out test tasks by context length. **(Right)** AMAGO enables larger models and longer context lengths than off-policy RNN variants. We compare training throughput in a common locomotion benchmark [97] with more details in Appendix D.

Figure 6 (left) breaks from our default context length $l = H$ and evaluates varying lengths on a sample of Meta-World ML-1 robotics environments [98]. Meta-World creates meta-RL tasks out of goal-conditioned environments by masking the goal information, which can be inferred from short

context lengths of dense reward signals. While the maximum meta-train performance of every context length is nearly identical, the meta-test success rates decrease slightly over long sequences. Extended contexts may encourage overfitting on these smaller environment distributions. Figure 6 (right) measures the scalability of AMAGO relative to off-policy agents with a separated RNN architecture as well as the more efficient (but previously unstable) shared model. AMAGO lets us train larger models faster than RNNs with equivalent memory capacity. More importantly, Transformers make optimizing these super-long sequences more realistic.

## 5.2 GOAL-CONDITIONED ENVIRONMENT ADAPTATION

We now turn our focus towards generalization over procedurally generated environments $e \sim p(e)$ and multi-step instructions $(g^0, \dots, g^k) \sim p(g \mid e)$ of up to $k$ goals (Sec. 3). A successful policy is able to adapt to a new environment in order to achieve sparse rewards of $+1$ for completing each step of its instruction, creating a general instruction-following agent. Learning to explore from sparse rewards is difficult, but AMAGO's long-horizon off-policy learning update lets us relabel trajectories with alternative instructions (Figure 2). Before we scale up to a more complex domain, we introduce two easily-simulated benchmarks. "Package Delivery" is a toy problem where agents navigate a sequence of forks in a road to deliver items to a list of addresses. This task is both sparse and memory-intensive, as achieving any reward requires recall of previous attempts. Appendix C.3 provides a full environment description, and Figure 7 compares key ablations of AMAGO. Relabeling, memory, and multi-gamma learning are essential to performance and highlight the importance of AMAGO's technical details.

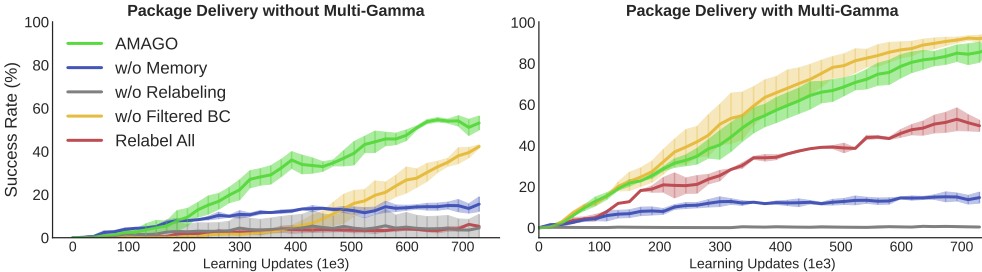

Figure 7: **Package Delivery Results.** Relabeling, long-term memory, and multi-gamma updates are essential to success in this sparse goal-conditioned adaptation problem ($H = 180$).

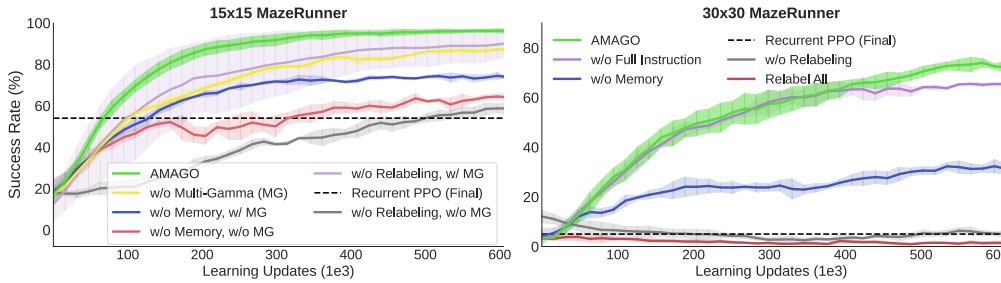

Figure 8: **MazeRunner Results.** Memory improves navigation in a partially observed maze, but sparse rewards create the most significant barrier to learning ($15 \times 15\ H = 400$, $30 \times 30\ H = 1,000$).

"MazeRunner" is a zero-shot maze navigation problem modeled after the example in Figure 2 and loosely based on Memory Maze [99]. The agent finds itself in a familiar spawn location in an otherwise unfamiliar maze and needs to navigate to a sequence of $k \in [1, 3]$ locations. Although the underlying maze map is generated as an $N \times N$ gridworld, the agent's observations are continuous Lidar-like depth sensors. When $N = 15$, the problem is sparse but solvable without relabeling thanks to our implementation's low-level improvements (Figure 8 left). However, $N = 30$ is impossibly sparse, and relabeling is crucial to success (Fig. 8 right). We run a similar experiment where action dynamics are randomly reset every episode in Appendix C.4. While this variant is more challenging,

our method outperforms other zero-shot meta-RL agents and nearly recovers the original performance while adapting to the new action space.

Crafter [33] is a research-friendly simplification of Minecraft designed to evaluate multi-task capabilities in procedurally generated worlds. Agents explore their surroundings to gather food and resources while progressing through a tech tree of advanced tools and avoiding dangerous enemies. We create an *instruction-conditioned* version where the agent is only successful if it completes the specified task. Our instructions are formed from the 22 original Crafter achievements with added goals for traveling to a grid of world coordinates and placing blocks in specific locations. Goals are represented as strings like "make stone pickaxe", which are tokenized at the word level with the motivation of enabling knowledge transfer between similar goals. Any sequence of $k \leq 5$ goal strings forms a valid instruction that we expect our agent to solve in any randomly generated Crafter environment.

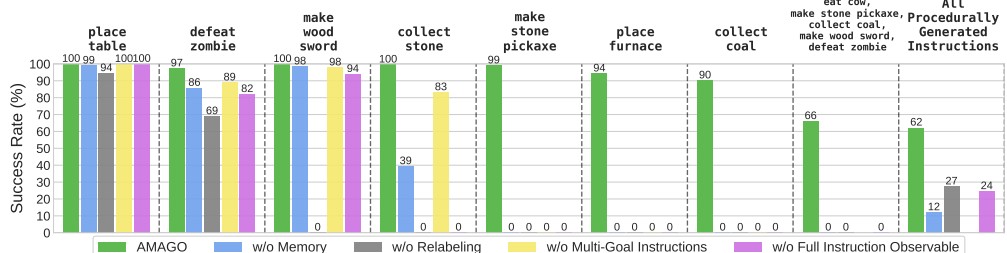

Figure 9: **Crafter Instruction Success Rates.** Here we highlight specific instructions that reveal exploration capabilities. "All Procedurally Generated Instructions" indicates performance across the entire range of multi-step goals in randomly generated worlds.

Figure 9 shows the success of AMAGO on the entire goal distribution along with several highlighted tasks. Resource-acquisition and tool-making tasks that require test-time exploration and adaptation to a new world benefit from Transformer policies and long-term memory. As the steps in an instruction become less likely to be achieved by random exploration, relabeling becomes essential to success. However, Crafter reveals a more interesting capability of our multi-step hindsight relabeling. If we ablate our relabeling scheme to single-goal HER and evaluate on instructions with length $k = 1$ (Fig. 9 "w/o Multi-Goal..."), we find AMAGO loses the ability to complete advanced goals. Crafter's achievement system locks skills behind a progression of prerequisite steps. For example, we can only make tools after we have collected the necessary materials. Relabeling with instructions lets us follow randomly generated sequences of goals we have mastered until we reach the "frontier" of skills we have discovered, where new goals have a realistic chance of occurring by random exploration. The agent eventually learns to complete new discoveries as part of other instructions, creating more opportunities for exploration. However, this effect only occurs when we provide the entire task in advance instead of revealing the instruction one step at a time (Fig. 9 "w/o Full Instruction..."). We continue this investigation in detail with additional Crafter experiments in Appendix C.5.

## 6 CONCLUSION

We have introduced AMAGO, an in-context RL agent for generalization, long-term memory, and meta-learning. Our work makes important technical contributions by finding a stable and high-performance way to train off-policy RL agents on top of one long-context Transformer. With AMAGO, we can train policies to adapt over more distant planning horizons with longer effective memories. We also show that the benefits of an off-policy in-context method can go beyond stability and sample efficiency, as AMAGO lets us relabel multi-step instructions to discover sparse rewards while adapting to unfamiliar environments. Our agent's efficiency on long input sequences creates an exciting direction for future work, as very few academic-scale meta-RL or RL generalization benchmarks genuinely require adaptation over thousands of timesteps. AMAGO is open-source, and the combination of its performance and flexibility should let it serve as a strong baseline in the development of new benchmarks in this area. The core AMAGO framework is also compatible with sequence models besides Transformers and creates an extensible way to research more experimental long-term memory architectures that can push adaptation horizons even further.

ACKNOWLEDGMENTS

This work was supported by NSF EFRI-2318065, Salesforce, and JP Morgan. We would like to thank Braham Snyder, Yifeng Zhu, Zhenyu Jiang, Soroush Nasiriany, Huihan Liu, Rutav Shah, Mingyo Seo, and the UT Austin Robot Perception and Learning Lab for constructive feedback on early drafts of this paper.

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

# A  AMAGO DETAILS

## A.1  SHARING A SINGLE SEQUENCE MODEL IN OFF-POLICY RL

Off-policy actor-critic methods compute loss terms with an ensemble of actor and critic networks along with a moving-average copy of their parameters used to generate temporal difference targets. Extending the feed-forward (fully observed) setup to sequence-based learning results in an excessive number of sequence model forward/backward passes per training step (Figure 10 top left) [69]. This has created several ways to share parameters and improve efficiency. Ni et al. [22] share a sequence model between the ensemble of critic networks, which becomes more important when using REDQ [74] (Fig. 10 top right). Parameters can be shared across the actor and critics, but this has been shown to be unstable. SAC+AE [72] confronts a similar problem in pixel-based learning and popularized the solution of detaching the larger base model (the Transformer in our case) from the actor's gradients (Fig. 10 bottom left). This approach has also been demonstrated in sequence-learning [71]. AMAGO removes the target sequence model as well — sharing one Transformer across every actor, critic, and target network while preserving the actor's gradients and training with one optimizer (Fig. 10 bottom right). Ni et al. [22] evaluate a fully shared architecture but find it to be unstable and do not consistently apply it across every domain. We find that instability is caused by the critic receiving gradients from the actor's loss and remove these terms during the backward pass of the joint actor-critic objective (Appendix A.2 Equation 3). Concurrent to our work, [96] addressed the same problem by using a frozen copy of the critics to compute the actor loss.

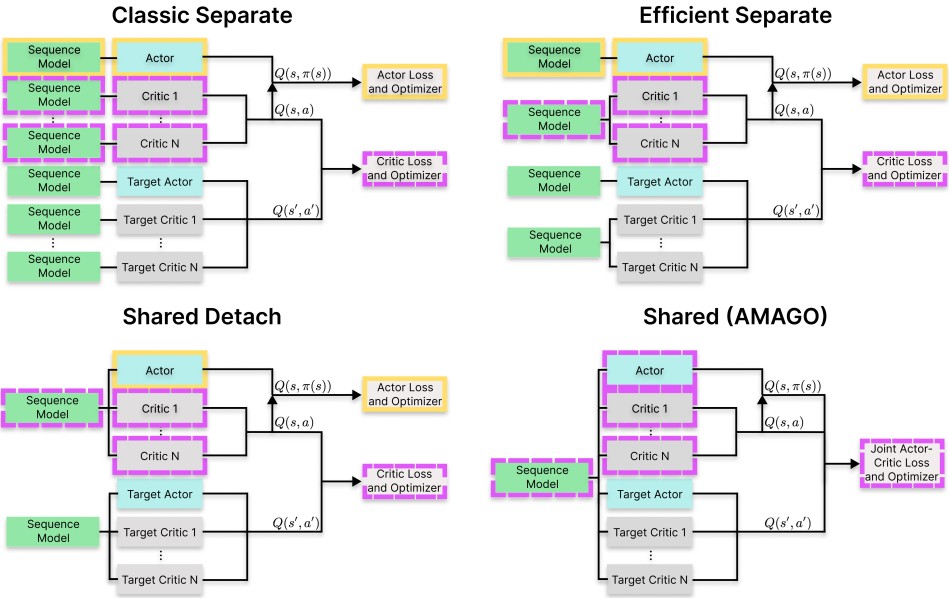

Figure 10: **Evolution of Off-Policy Actor-Critic Agent Architectures.** Black arrows give a high-level overview of how the actor, critic(s), and target networks combine to compute the training objective(s). Network borders are color-coded according to the loss function they optimize. Sequence models (green) are more expensive than feed-forward actors (blue) and critics (gray), which has motivated several ways to simplify the training process while maintaining stability.

## A.2  BASE ACTOR-CRITIC UPDATE

AMAGO's shared sequence model reduces the RL training process to the standard feed-forward case where the output of the trajectory encoder becomes the state array ($s$), and the batch size is effectively larger by a factor of the context length $l$. At a high level, we are training a stochastic policy $\pi$ with a custom variant of the off-policy actor-critic update derived from DDPG [69]. In continuous control, the critic $Q$ takes actions as a network input and outputs a scalar value. In discrete environments, the critic outputs an array corresponding to the value for each of the $|\mathcal{A}|$ actions [70]. The actor is trained

to maximize the output of the critic ($\mathcal{L}_{\text{PG}}$) while the critic is trained to minimize the classic one-step temporal difference error ($\mathcal{L}_{\text{TD}}$):

$$\mathcal{L}_{\text{PG}}(s) = -Q(\cancel{\nabla}s, \pi(s)) \qquad\qquad \text{Actor Term} \qquad (1)$$

$$\mathcal{L}_{\text{TD}}(s, a, r, s') = \big(Q(s,a) - (r + \gamma\cancel{\nabla}\bar{Q}(s', \bar{\pi}(s'))))\big)^2 \qquad \text{Critic Term} \qquad (2)$$

Where $\cancel{\nabla}$ is a stop-gradient, and $\bar{Q}$ and $\bar{\pi}$ denote target critic and actor networks, respectively. AMAGO then combines these two terms into a single shared loss:

$$\mathcal{L}_{\text{AMAGO}} = \underset{\tau \sim \mathcal{D}}{\mathbb{E}} \left[ \frac{1}{l} \sum_{t=0}^{l} \lambda_0 \mathcal{L}_{\text{TD}}(s_t, a_t, r_t, s_{t+1}) + \lambda_1 \mathcal{L}_{\text{PG}}(s_t) \right] \qquad (3)$$

As mentioned in Appendix A.1, we zero gradients to prevent our critic from directly minimizing the actor's objective $\mathcal{L}_{\text{PG}}$. The weights of each term $(\lambda_0, \lambda_1)$ can be important but unintuitive hyperparameters. $\mathcal{L}_{\text{PG}}$ and $\mathcal{L}_{\text{TD}}$ do not scale equally with $Q$, and the scale of $Q$ values depends on the environment's reward function and changes over time at a rate determined by learning progress. This means that the relative importance of our loss terms to their shared trajectory encoder's gradient update is shifting unpredictably, making $(\lambda_0, \lambda_1)$ difficult to set in a new environment.

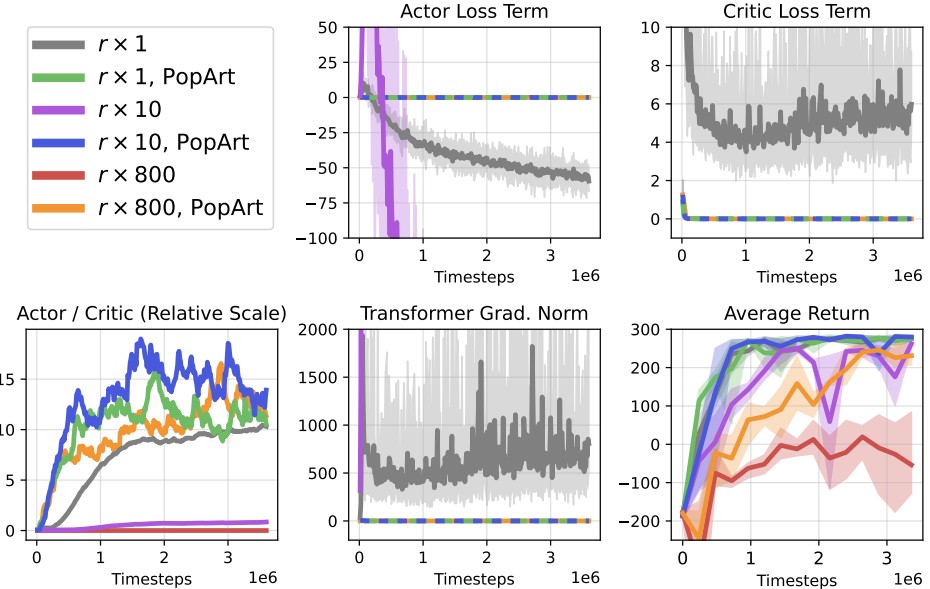

Figure 11: **Scaling AMAGO's Learning Update with PopArt.** PopArt automatically places the relative importance of the actor and critic loss terms in AMAGO's shared learning update on a reasonably predictable scale, and enables stable training without extreme gradient clipping.

PopArt [85] is typically used to normalize the scale of value-based loss functions when training one policy across multiple domains. However, we use it to reduce hyperparameter tuning by putting $Q$ (and therefore $\mathcal{L}_{\text{PG}}$ and $\mathcal{L}_{\text{TD}}$) on a predictable scale so that $(\lambda_0, \lambda_1)$ can have meaningful default values. Figure 11 demonstrates the problem and PopArt's solution. In this example, we train context length $l = 128$ AMAGO agents on LunarLander-v2 [100]. We use our default values $(\lambda_0, \lambda_1) = (10, 1)$ (Table 4). The gray curves track the actor and critic objectives on the environment's default reward scale without using PopArt. Performance happens to be quite strong (Fig. 11 lower right), but the actor and critic loss scales make gradient norms (lower center) destructively large without clipping. When we scale rewards by a constant ($r \times 10$, $r \times 800$), the optimization metrics often cannot be shown on a readable y-axis and the relative importance of the actor term is now nearly zero (lower left). PopArt automatically puts the relative importance of the actor loss on a predictable order of magnitude (blue, green, orange), and we are no longer relying on alarming levels of gradient clipping.

**Critic Ensembling.**    In practice, the actor's goal of maximizing the critic's output leads to value overestimation that is handled by using the minimum prediction of two critics trained in parallel [73]. Overestimation is especially concerning in our case, because AMAGO's use of long sequences means that its effective replay ratio [79] can be unusually high; it is not uncommon for our agents to train on their entire replay buffer several times between rollouts. We enable the use of REDQ ensembles [74] with more than two critics as a precaution.

**Filtered Behavioral Cloning.**    One difference between training policies with a supervised objective (where Transformers are common and relatively stable) and RL is that our actor's update depends on the scale and stability of the $Q$-value surface (Eq. 1). We can improve learning with a "filtered" behavior cloning (BC) actor objective that is independent of the scale of the critics' output space, which is added to Eq. 3 with a third weight $\lambda_2$:

$$A(s,a) = Q(s,a) - V(s) = Q(s,a) - \mathbb{E}_{a' \sim \pi(s)}[Q(s,a')] \qquad \text{Advantage Estimate} \quad (4)$$

$$f(s,a) = \mathbb{1}_{\{A(s,a)>0\}} \qquad\qquad\qquad\qquad\qquad \text{Binary Filter [92]} \quad (5)$$

$$\mathcal{L}_{\text{FBC}}(s,a) = -f(s,a)\log\pi(a \mid s) \qquad\qquad\qquad \text{Filtered BC Term} \quad (6)$$

The actor's standard $\mathcal{L}_{\text{PG}}$ term has a strong learning signal when the value surface is steep, but $\mathcal{L}_{\text{FBC}}$ only depends on the sign of the advantage and behaves more like supervised learning. AMAGO is now always optimizing a stable sequence modeling objective where we learn to predict replay buffer actions with positive advantage. Variants of the filtered BC update appear in online RL but have become more common in offline settings [101, 102]. AMAGO's gradient flow causes $\lambda_2 \mathcal{L}_{\text{FBC}}$ to impact the objective of the trajectory encoder and actor network but not the critic. We use the binary filter from CRR [92] because it does not add hyperparameters, and training on batches of long sequences helps mask its tendency to increase variance by filtering too many actions [103].

**Multi-Gamma Learning.**    Long horizons and sparse rewards can lead to flat $Q$-value surfaces and slow the convergence of TD learning. AMAGO computes $\mathcal{L}_{\text{PG}}$, $\mathcal{L}_{\text{TD}}$, and $\mathcal{L}_{\text{FBC}}$ in parallel across $\gamma_N$ values of the discount factor $\gamma$. Each $\gamma$ creates its own value surface — informally making it less likely that all of our loss terms have converged and improving representation learning of shared parameters. Discrete actor and critics' output layers become $\gamma_N$ times larger. Actions for each $\gamma$ need to be an input to continuous critics (along with $\gamma$ values themselves), so the effective batch size of continuous-action critic networks is multiplied by $\gamma_N$. In either case, however, the relative cost of this technique becomes low as the size of the shared Transformer increases. An example of the $Q$ scales learned by different values of $\gamma$ is plotted in Figure 12. During rollouts, we can select the index of the actor's outputs corresponding to any of the $\gamma_N$ horizons used during training. This selection could potentially be randomized to generate more diverse behavior, but we select a fixed $\gamma = .999$ in our experiments.

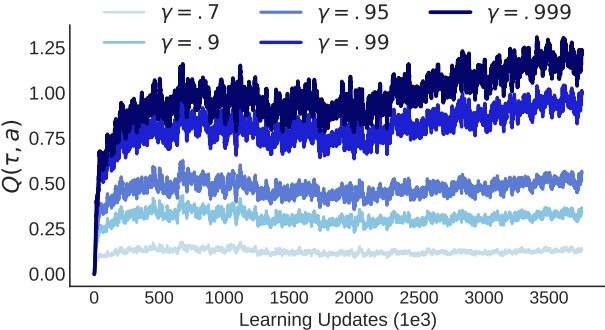

Figure 12: **Multi-Gamma Actor-Critic Training.** AMAGO optimizes one Transformer on actor-critic loss terms corresponding to many values of the discount factor $\gamma$, and can use the actions that maximize any horizon at test-time. We show the average Q-value across trajectory sequences throughout training at different discounts.

Our multi-gamma update is motivated by a need to improve learning signal for unstable sequence models in long-horizon actor-critic updates with discrete and continuous actions. This approach was developed as a natural extension of a training objective that is already parallelized across timesteps and an ensemble of critics. After the initial release of our work, we learned that a discrete value-based [3, 104] version of the multi-gamma update has previously been studied as a byproduct of hyperbolic discounting in pixel-based MDP environments [91, 105]. The hyperbolic discounting perspective actually motivates a more diverse range of $\gamma$ values than used in our results (Table 4), and may greatly improve AMAGO's sample efficiency (Appendix C.1 Figure 18).

**Stochastic Policies and Exploration.** AMAGO samples from a stochastic policy during both data collection and evaluation. Because we do not use entropy constraints [106], we add exploration noise during data collection. In discrete domains, noise is added as in classic epsilon-greedy, while continuous domains randomly perturb the action vector as in TD3 [73]. The level of action randomness is determined by a hyperparameter $\epsilon$ that is typically annealed over a fixed number of environment steps. AMAGO adapts this schedule to more closely align with the exploration/exploitation trade-off that occurs within the adaption horizon of any given CMDP [64]. $\epsilon$ is annealed over the $H$ timesteps of a rollout, and the intensity of this schedule decreases over training. This process is visualized in Figure 13. Because the parameters of this schedule are difficult to tune, we heavily randomize them across AMAGO's parallel actors — meaning we are always collecting data at varying levels of action noise. Due to this randomization there are no main experiments where we found tuning the exploration schedule to be critical to achieving strong results[3]. In fact, randomizing over $\epsilon$ is probably the more important implementation detail, but we describe the rest of the approach for completeness. The slope of the episode-level schedule can be set to zero to recover the standard approach.

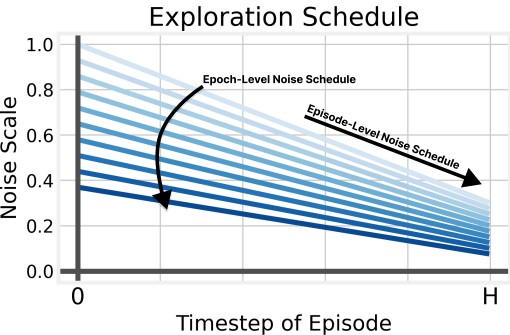

Figure 13: **Exploration Noise in Implicit POMDPs.** AMAGO adapts the standard random exploration schedule to more closely align with the exploration/exploitation trade-off that occurs when acting in an implicit POMDP.

### A.3 AMAGO ARCHITECTURE

**Transformer.** We observe performance collapse when using a standard Pre-LayerNorm [77] Transformer in long training runs. In rare cases, we find that this is caused by gradient collapse due to saturating ReLU activations. For this reason we replace every ReLU/GeLU (including those in the actor/critic MLPs) with a Leaky ReLU that will allow learning to continue. This idea is also motivated by work in network plasticity and long training runs in continual RL, where activations other than ReLU can be a simple baseline [83]. We find that this change fixes gradient instability, but does not prevent performance collapse. Instead, collapse is now caused by saturating activations in the residual block of AMAGO's Transformer. We apply two existing methods that effectively solve this problem. Normformer's [88] additional LayerNorms [107] isolate the optimization problem to the query/key/value activations whose saturation directly causes attention entropy collapse. $\sigma$Reparam [87] stabilizes attention by limiting the magnitude of queries, keys, and values. Figure 14 demonstrates this pattern of activations on a sample POPGym environment where the optimal policy requires recall of a specific timestep and encourages low-entropy attention matrices. However, we observe

---

[3]The toy T-Maze memory result (Figure 5) uses a unique schedule discussed in Appendix C.2, but this adjustment is motivated by the environment setup and is not based on tuning.

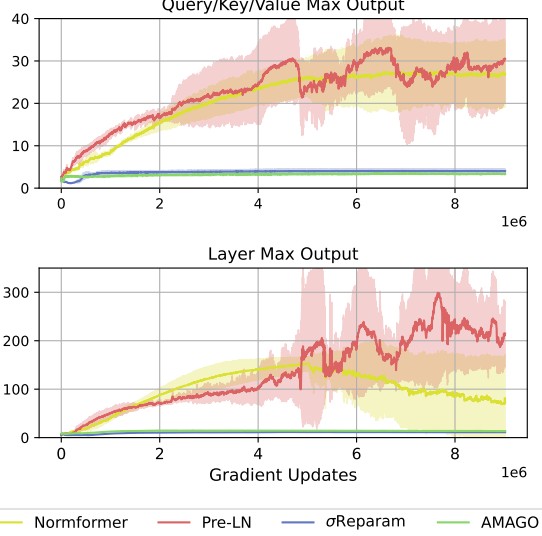

Figure 14: **Transformer Residual Block Activations in a Low-Entropy Attention Environment.** We record the maximum output of a Transformer layer and its query/key/value vectors in our default POPGym architecture while training in a recall-intensive environment where the optimal policy encourages a low-entropy attention matrix (Figure 15).

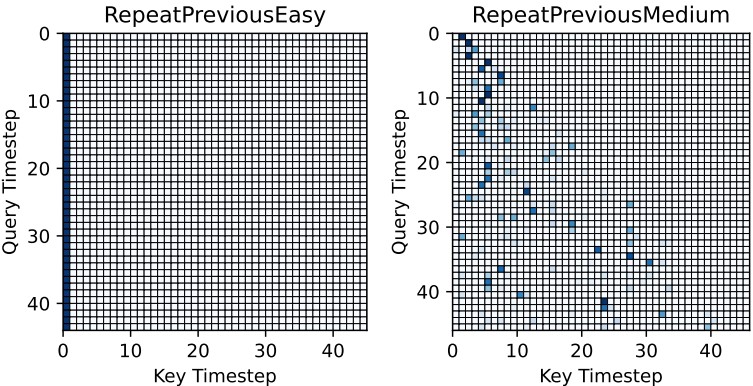

Figure 15: **Examples of Low-Entropy Attention Matrices.** We visualize representative examples of AMAGO attention heads in two recall-intensive POPGym environments on the 45th timestep of a rollout (for readability). Darker blue entries indicate high attention weights. Both policies are nearly optimal with average returns $> .99$.

collapse due to saturating activations in many environments in our experiments if training continues for long enough — even when performance has not yet converged. Our architectural changes let us stably train sparse attention patterns like those visualized in Figure 15. AMAGO uses Flash Attention [108] to enable long context lengths on a single GPU (Figure 29). Figure 16 summarizes our default architectural changes.

**Instruction-Conditioning.** AMAGO uses a small RNN or MLP to process the instruction sequence of goal tokens, and the resulting representation is concatenated to the CMDP information that forms Transformer input tokens. It would be simpler to add the instruction to the beginning of the context sequence. The only reason for the extra complexity of the goal embedding is to allow for fair baselines that do not use context sequences ("w/o Memory").

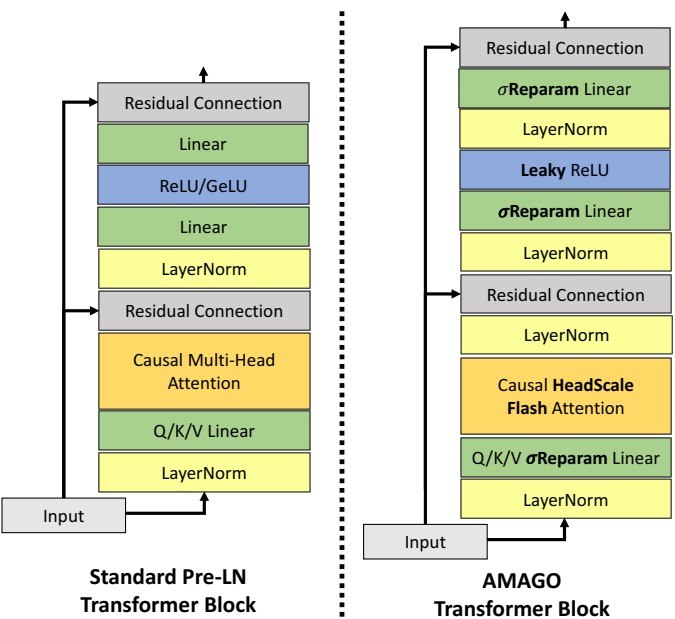

Figure 16: **AMAGO Transformer Block. (Left)** A standard Pre-LayerNorm (Pre-LN) Transformer layer [77]. **(Right)** AMAGO replaces all saturating activations with Leaky ReLUs and uses additional LayerNorms [107] (as in NormFormer [88]) and a modified linear layer ($\sigma$Reparam [87]). These strategies limit the magnitude of activations along the residual block and effectively prevent attention entropy collapse.

## B  RELABELING WITH GOAL IMPORTANCE SAMPLING

AMAGO generates training data in multi-goal domains by relabeling trajectories with alternative instructions based on hindsight outcomes. Relabeling improves reward sparsity for actor-critic training, and greatly amplifies the learning signal of existing data by recycling the same experience with many different instructions. This technique works by saving the rewards for the entire goal space during rollouts, rather than just the rewards for the goals in the intended instruction (Figure 17 Step 1). While evaluating many different dense reward terms would be unrealistic, it is more practical in sparse goal-conditioned domains where success can be evaluated with simple rules. Algorithm 1 provides a high-level overview of multi-step relabeling. This technique reduces to HER [45] when: 1) the goal space is a subset of the state space, 2) goal sequence lengths $k = 1$, and 3) alternative goals are primarily sampled from the end of the trajectory (Alg. 1 line 4).

---

**Algorithm 1** Simplified Hindsight Instruction Relabeling

---

**Require:** Trajectory $\tau$ with goal sequence $g = (g^0, \ldots, g^k)$ of length $k$

1: $n \leftarrow$ number of steps in $g$ successfully completed by $\tau$
2: $(t_{g^0}, \ldots, t_{g^n}) \leftarrow$ timesteps where each sub-goal of $g$ was achieved
3: $h \leftarrow$ relabel_count$(0, k - n) \in [0, k - n]$ ▷ Choose a number of hindsight goals to insert. Defaults to uniform sampling.
4: $(a^0, \ldots, a^h), (t_{a^0}, \ldots, t_{a^h}) \leftarrow$ sample_alternative_goals$(\tau)$ ▷ Sample a goal from $h$ timesteps in $\tau$ that completed alternative objectives. Defualts to uniform sampling.
5: $r \leftarrow$ sort$\big((a^0, \ldots, a^h, g^0, \ldots, g^n), \text{by=}(t_{a^0}, \ldots, t_{a^h}, t_{g^0}, \ldots, t_{g^n})\big)$ ▷ Insert new goals in chronological order.
6: $\tau' \leftarrow$ replay$(\tau, r)$ ▷ Recompute rewards and terminals based on goal sequence $r$ (Fig. 2).

---

Generating a diverse training dataset with relabeled sequences of goals allows our agents to carry out multi-stage tasks and has important exploration advantages (Appendix C.5), but creates a practical

issue where we have too many alternative instructions to choose from. Domains like Crafter and MazeRunner create rollouts with dozens or hundreds of candidate goals over the full length of the trajectory. There are also goal types that can occur simultaneously and for many consecutive timesteps. AMAGO relabels by sampling one instruction from the many sub-sequences of these goals (Alg. 1 line 4). With so many combinations of goal instructions available to us, we need a way to focus our learning updates on useful information.

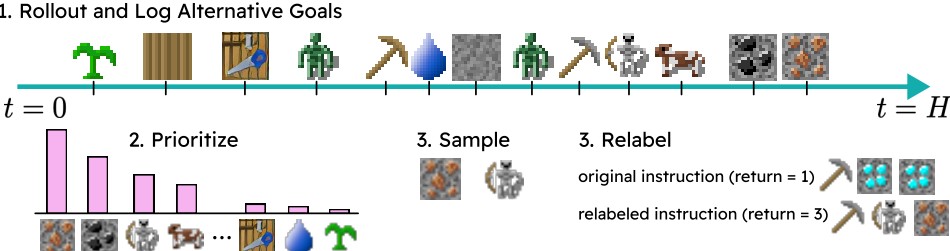

Figure 17: **Relabeling With Prioritized Goal Sampling.** Long rollouts in multi-goal domains lead to an unmanageable number of candidate instructions for relabeling. AMAGO improves sample efficiency without domain knowledge by prioritizing rare goals.

Our solution is a weighted relabeling scheme that helps sort through the noise of common outcomes by prioritizing interesting goals. While there could be opportunities to add domain knowledge in this process, we prefer to avoid this and sample goals according to their rarity. AMAGO tracks both the frequency that a particular goal occurs at any given timestep, and the frequency that it occurs at all in a given episode. We assign a priority score to goals based on their rarity, which then lets us modify the relabeling scheme to sample based on these scores (Figure 17). AMAGO's technical details are designed to reduce hyperparameter sensitivity and prevent individual tricks like this from becoming unintuitive points of failure that require manual tuning. Therefore we automatically randomize over several reasonable approaches. Examples include sampling from the top-$k$ most rare goals according to either frequency statistic, or those above either the median or minimum rarity in a trajectory to filter trivial goals. AMAGO still relabels uniformly with some frequency, which keeps the full diversity of our dataset available and prevents information from being lost. Randomization over these implementation details occurs on a per-trajectory level, meaning every batch has sequences that were generated with a wide range of strategies. We defer the precise details to our open-source code release. Appendix C.5 provides a quantitative demonstration of our method.

## C EXPERIMENTAL DETAILS AND FURTHER ANALYSIS

This section provides a detailed analysis of AMAGO's main results. Each subsection will give a description of our custom learning domains, followed by a more complete discussion of the results in the main text with additional experiments. Hyperparameter and compute information is listed in Appendix D.

### C.1 POPGYM

We evaluate on 39 environments from the POPGym suite [30] and follow the original benchmark in using policy architectures with a memory hidden state of 256. Learning curves for each environment are shown on a full page below. Figure 3 reports results at the benchmark standard of 15M timesteps, though the learning curves extend slightly further when data is available. We plot the maximum and minimum value achieved by any seed to highlight the stability of AMAGO relative to the "naive" off-policy Transformer baseline. Each environment defaults to 3 random seeds, as in [30]. The variance of AMAGO at convergence is extremely low. However, there can be significant variance in the timestep where rapid increases in performance begin. There are two environments where this variance impacts results because it occurs near the 15M sample limit (AutoencodeHard and CountRecallHard), which we address by running 9 random seeds.

The "naive" agent maintains AMAGO's shared actor-critic update but disables many of our other technical details. These include the modified Transformer architecture (Appendix A.3) and multi-

gamma update. The naive agent also reduces the REDQ ensemble from AMAGO's default of 4 parallel critics to the standard 2, and uses a lower discount factor of $\gamma = .99$. This causes learning to collapse in many seeds. However, collapse generally does not impact the final scores reported in Figure 3 which indicate the *maximum mean episodic return* (MMER) achieved during training. The effects of policy collapse can be much more damaging in longer goal-conditioned experiments where it occurs before convergence, but is too expensive to demonstrate at this scale.

AMAGO was briefly tuned on one environment (ConcentrationEasy) to meet the benchmark's sample limit. However, our results suggest that we likely did not push sample efficiency settings high enough, as AMAGO is still improving well after 15M timesteps in some difficult environments. The multi-gamma update (Appendix A.2) provides a useful starting point for increased sample efficiency. The combination of this techniques' compute efficiency and our focus on long-horizon learning motivated a large number of $\gamma > .99$ (Table 4) throughout our work. However, another motivation for a similar update inspires a broader range including low $\gamma$ values [91, 105]. We experiment with a wider range of settings in two POPGym environments where our main results are limited by sample efficiency in Figure 18.

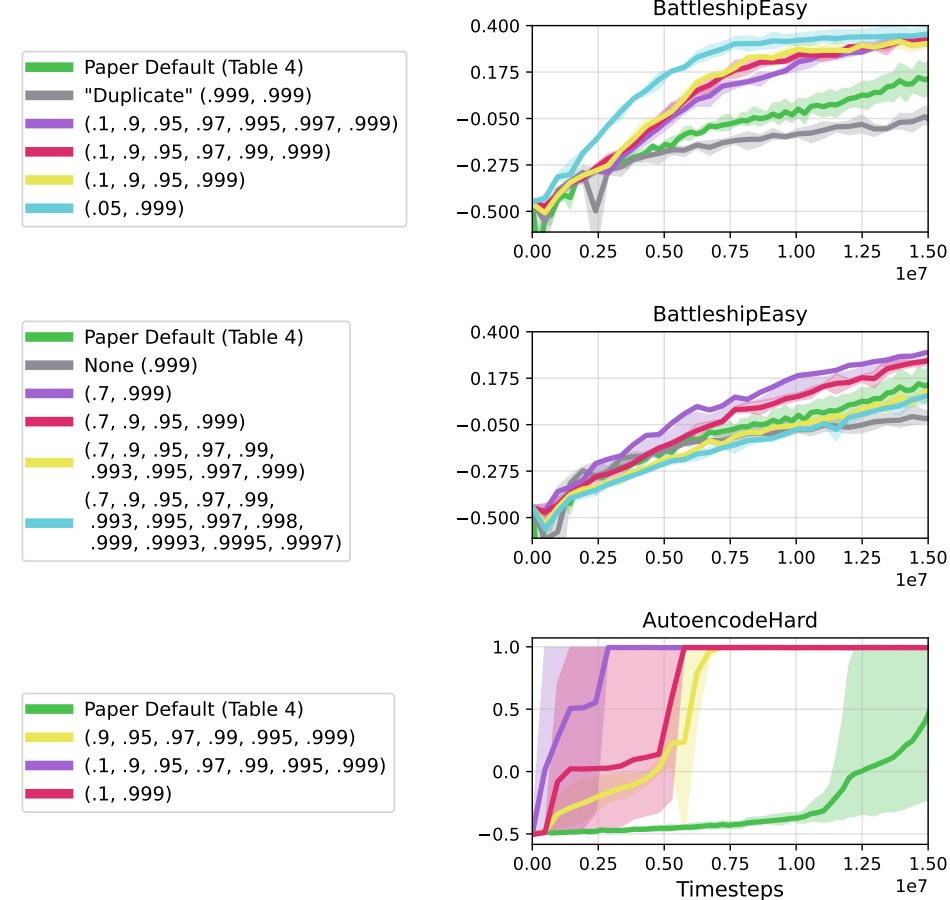

Figure 18: **Alternative Multi-Gamma Settings in POPGym.** We evaluate a range of settings for the discount factor ensemble used in AMAGO's learning update, and find that the defaults used in our main experiments may be underestimating the importance of short-horizon $\gamma$ values.

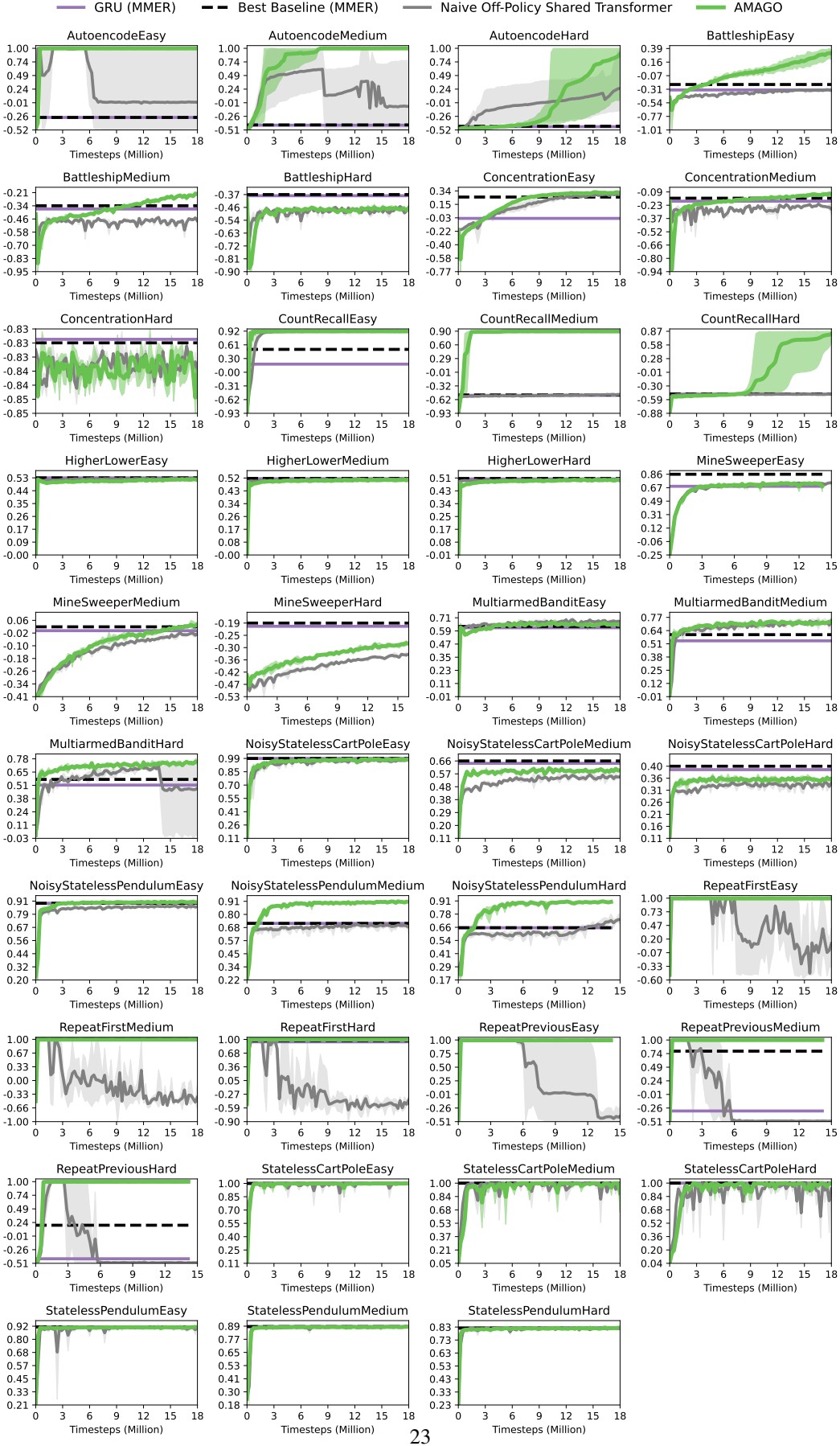

## C.2 ADDITIONAL MEMORY AND MULTI-EPISODIC META-RL RESULTS

**Half-Cheetah Velocity.** In Figure 5 (left) we use a classic meta-RL benchmark from [97] as an example of a case where AMAGO's long sequences and high discount factors are clearly unnecessary but do not need tuning. This task is solvable with short context lengths of just a few timesteps. We evaluate adaptation over the first 3 episodes ($H = 600$) and report baselines from variBAD [64], BOReL [109], RL$^2$ [11], and the recurrent off-policy implementation in [22].

**Dark-Key-To-Door.** Dark-Key-To-Door is a sparse-reward multi-episodic meta-RL problem [28]. Figure 19 compares a version of AMAGO with a Transformer and RNN trajectory encoder on a $9 \times 9$ version of the environment with a max episode length of 50. An agent that always fails to solve the task will encounter 10 episodes of a new environment during a meta-testing horizon of $H = 500$ timesteps. The maximum return per episode is 2. An adaptive agent learns to solve the task quickly once it has identified the key and door location by meta-exploration, and tries to complete as many episodes as possible in the $H = 500$ timestep limit. The Transformer and RNN model architectures have equal memory hidden state size and layer count, and all other training details are held fixed.

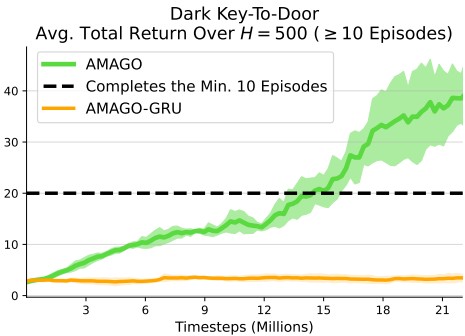

Figure 19: **Dark Key-To-Door Trajectory Encoder Comparison.** On-Policy RNNs have previously been successful in this environment but fail to make progress at $l = H = 500$ when directly substituted into the AMAGO agent.

We replicate Algorithm Distillation [28] (AD) on an $8 \times 8$ version of the task by collecting training histories from 700 source RL agents. Each history is generated by a memory-free actor-critic as it improves for $50,000$ timesteps in a fixed environment $e$. We then train a standard Transformer architecture on the supervised task of predicting the next action in sequences sampled from these learning histories. While AD converges to high performance, it does so on roughly the same timescale as the single-task agents in its training data (Figure 20 right). AMAGO can directly optimize for fast adaptation over the given horizon $H$, and learns a much more efficient strategy over the first few episodes in a new environment (Fig. 20 left).

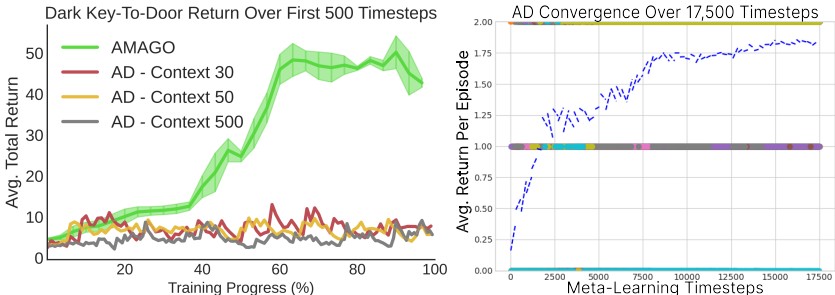

Figure 20: **Fast-Adaptation with Long-Context Transformers.** AMAGO uses a long-context Transformer to directly optimize for fast adaptation to new environments **(Left)**, while AD [28] converges at approximately the speed of the single-task agents that generated its training data **(Right)**.

**Passive T-Maze.**    Ni et al. [96] is a concurrent work that includes a T-maze experiment to effectively unit-test the recall ability of in-context agents; any policy that achieves the maximum return must be able to recall information from the first timestep at the final timestep $H$. Their results show that recurrent approaches fail around context lengths of 100, but that Transformers can stretch as high as $H = 1,500$. The T-Maze task is intended to isolate memory from the effects of credit assignment and exploration. However, at extreme sequence lengths the reward begins to test the sample efficiency of epsilon-greedy exploration because reaching the memory challenge requires following the optimal policy for at least the first $H - 1$ timesteps. The exploration noise schedule (Appendix A) is modified to have standard noise near the beginning and end of an episode (when there are interesting actions to learn) but low noise in-between when it would waste a rollout to deviate from the learned policy. We avoid sampling from our stochastic policy because numerical stability protections prevent it from becoming fully deterministic, and this error accumulates over long rollouts. The reward penalty is also adapted to remain informative after the first timestep the policy disagrees with the optimal policy. These changes are not important to the main question of long-term recall but allow this environment to evaluate long sequence lengths in practice. With these adjustments, AMAGO can stably learn the optimal policy all the way out to the GPU memory barrier with context lengths $l = H = 10,000$ (Figure 5 top right). However, as noted in [96] our choice of sequence model has no effect on the credit assignment problem of RL backups and AMAGO's Transformer does not lead to a significant improvement in the "Active T-Maze" variant of this problem.

**Wind Environment.**    Figure 5 (bottom right) reports scores of a sparse-reward meta-RL task from [22]. This task has continuous actions and is noted to be sensitive to hyperparameters. We evaluate AMAGO with a Transformer and RNN trajectory encoder — keeping all other details the same as in the Dark-Key-To-Door experiment. The RNN encoder is much more effective at the shorter sequence lengths in this environment ($H = 75$) than the Dark-Key-To-Door environment. It appears that the RNN is slightly more sample efficient than the Transformer when using the same update-to-data ratio.

**Meta-World ML-1.**    We train AMAGO for up to 10M timesteps with various context lengths (Figure 6 left). The checkpoint that reached the highest average return on the training tasks is evaluated on held-out test tasks. Each episode lasts for 500 timesteps and we evaluate for less than the 10 trials used in the original results [98] because we notice the success rates saturate within the first attempt. We have also experimented with the ML-10 and ML-45 multi-task variants of Meta-World. Preliminary results suggest that short context lengths are still surprisingly capable of identifying the task. However, we find that success in these benchmarks is very dependent on the agent's ability to independently normalize the scale of rewards in each of the 10 or 45 domains. This issue is not specific to meta-RL and is common in multi-task RL generally [85, 110]. AMAGO assumes we do not know how many tasks make up $p(e)$ (because in most of our other experiments this number is very large), so we leave this to future work.

## C.3    Package Delivery

Package Delivery is a toy problem designed to test our central focus of learning goal-conditioned policies that need to adapt to a distribution of different environments. The agent travels along a road and is rewarded for making deliveries at provided locations along the way. However, there are "forks" in the road where it needs to decide whether to turn left or right. If the agent picks the wrong direction at any fork, or reaches the end of the road, it is sent back to the starting position. The correct approach is to advance to a fork in the road, pick a direction, note whether it was the correct choice, and then recall that outcome the next time we reach the same decision. For an added factor of variation, the action that needs to be taken to be rewarded for dropping off a package is randomized in each environment, and needs to discovered by trial-and-error and then remembered so we do not run out of packages.

### C.3.1    Environment and Task Details

**Environment Distribution** $p(e)$: we randomly choose $f \sim U(2, 6)$ locations for forks on a road with length $L = 30$. Each fork is uniformly assigned a correct direction of left or right. These hyperparmeters could be tuned to create more difficult versions of this problem. We also uniformly sample a correct action for package delivery from 4 possibilities.

**Goal Space** $\mathcal{G}$: all of the locations $(0, \ldots, L = 30)$ where packages could be delivered.

**Observation Space**: current position, remaining packages, and a binary indicator of arriving at a fork in the road.

**Action Space**: 8 discrete actions corresponding to moving forwards along the road, left/right at a fork, standing still (no-op), and the 4 actions that could correspond to successfully dropping off a package (depending on the environment $e$).

**Instruction Distribution** $p(g \mid e)$: we randomly choose $k \sim U(2, 4)$ locations (without replacement) that are not road forks for package delivery.

**Task Horizon** $H$: This is the shortest problem in our goal-conditioned experiments with a maximum horizon (and therefore a maximum sequence length) of $H = 180$.

### C.3.2 ADDITIONAL RESULTS AND ANALYSIS

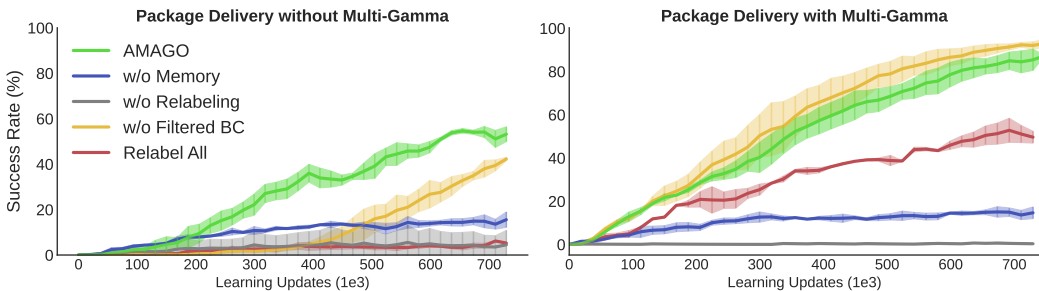

**Figure 7: Delivery Results** (reproduced here for convenience).

Figure 7 compares ablations of AMAGO on the Package Delivery domain. The main result is that AMAGO can successfully learn the memory-based strategy necessary to adapt to new roads. As expected, removing its context-based meta-learning capabilities ("w/o Memory") greatly reduces performance. The memory challenge of generalizing to new environments makes rewards difficult to find and relabeling is essential to learning any non-random behavior ("w/o Relabeling"). This gap between relabeling and not relabeling is a common theme across all our goal-conditioned experiments, and causes most external baselines to fail in an uninformative way. Figure 7 directly compares ablations with and without multi-gamma learning. Multi-gamma is one of several implementation details in AMAGO meant to enable stable learning from sparse rewards with Transformers. Filtered behavior cloning ("w/o Filtered BC") [92] is also meant to address sparsity, but we can see that its impact goes from quite positive (Fig. 7 left) to slightly negative (Fig. 7 right) when combined with multi-gamma. Filtered BC and multi-gamma are addressing a similar problem, but multi-gamma does so more effectively and exposes some of the drawbacks of filtered BC once the sparsity issue is resolved. For example, filtered BC can lead to a higher-entropy policy by cloning sub-optimal actions that leak through a noisy filter.

AMAGO randomly relabels trajectories to a mixture of returns between the true outcome and a complete success. The "Relabel All" curves follow the goal-conditioned supervised learning (GCSL) (Sec. 2) approach of relabeling every trajectory with a successful instruction. While this may improve data quality, it turns the value learning problem from predicting *if* a goal will be achieved to *when* it will happen, because from the agent's perspective every goal always does [60]. AMAGO's high discount factors cause this to have low learning signal, making actor optimization difficult. Multi-gamma improves performance (Fig. 7 right), because many of the $\gamma$ values used during training have a short enough horizon to mask the problem. Relabeling all trajectories to be a success while turning off the BC filter and policy gradient actor loss would create a GCSL method, which have shown promise with large Transformer architectures (Sec. 2). However we were unable to achieve competitive performance with this setup and prefer a more traditional RL learning update.

### C.4 MAZERUNNER

MazeRunner is a difficult but easily simulated problem that combines sparse goal-conditioning with long-term exploration. The agent needs to learn to navigate a randomly generated maze in order to reach a sequence of goal coordinates.

#### C.4.1 ENVIRONMENT AND TASK DETAILS

**Environment Distribution** $p(e)$: we randomly generate an $N \times N$ maze, and then manually adjust the bottom three rows to have a familiar layout where the agent starts in a small tunnel with open space on either side. An example $30 \times 30$ environment is rendered in Figure 21. We can optionally generate a random permutation of the action space, which adjusts the actions that corresponds to each direction of movement.

**Goal Space** $\mathcal{G}$: all of the locations $((0,0), \ldots, (N, N))$ in the maze.

**Observation Space**: 4 depth sensors from the agent's location to the nearest wall in each direction. By default we include the $(x, y)$ coordinates of the agent's position, which can be removed for an extra challenge that forces the agent to self-localize.

**Action Space**: Either 4 discrete actions or a 2-dimensional continuous space that is mapped to the 4 cardinal directions. The action directions can be randomized based on the environment $e$.

**Instruction Distribution** $p(g \mid e)$: we choose $k \sim U(1, 3)$ locations (without replacement) that are not covered by a wall of the maze.

**Task Horizon** $H$: The task horizon creates a difficult trade-off between exploration and exploitation. There can also be worst-case scenarios where the layout of the random maze and position of the goals make completing the task impossible in a fixed time limit. We compute difficulty statistics with an oracle tree-search agent and use the results to pick safe values where the $15 \times 15$ version sets $H = 400$ while $30 \times 30$ sets $H = 1,000$.

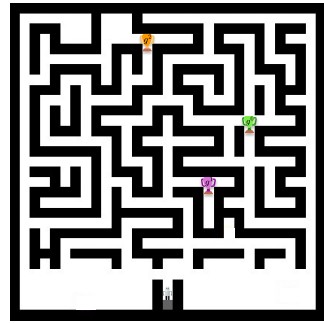

Figure 21: **$30 \times 30$ MazeRunner.** The agent begins in the bottom center with goal locations shown as colored trophies.

#### C.4.2 ADDITIONAL RESULTS AND ANALYSIS

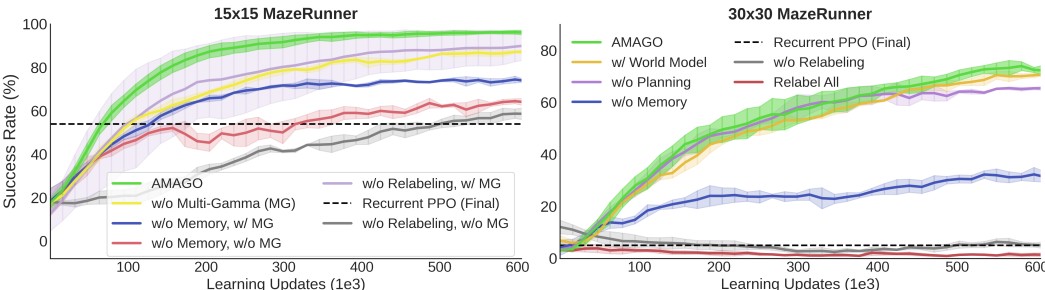

**Figure 8: MazeRunner Results** (reproduced here for convenience).

**$15 \times 15$ MazeRunner.** Figure 8 (left) evaluates the default version of MazeRunner with $N = 15$. Without random actions dynamics and with the current status of the task as an input to each timestep, a standard Recurrent PPO baseline could learn the same kind of adaptive generalization as AMAGO. The lowest-feature ablation of AMAGO's core agent ("w/o Relabeling, w/o Multi-Gamma") uses a Transformer to match the performance of Recurrent PPO on the $15 \times 15$ domain. However, our other technical improvements can almost double that performance, albeit with high variance ("w/o Relabeling, w/ Multi-Gamma"). Adding relabeling to create the full AMAGO method leads to a nearly perfect success rate in $15 \times 15$ with low variance ("AMAGO"). Position information $(x, y)$ was included to help baseline performance, but we find that AMAGO can achieve a $91\%$ success rate without it.

**30 × 30 MazeRunner.** The larger maze size creates an extremely challenging sparse exploration problem, and all ablations of the relabeling scheme fail along with the Recurrent PPO baseline. AMAGO performs well and uses its memory to efficiently explore its surroundings ("w/o Memory" has about half its final success rate). We believe there is room to continue to use the MazeRunner environment at larger maze sizes and task horizons to evaluate long-term-memory architectures at low simulation cost. We experiment with adding a dynamics modeling term to AMAGO's training objective ("w/ World Model"). World modeling is a natural fit for AMAGO's long-context Transformer backbone but does not appear to have an impact on this low-dimensional domain. The "w/o Planning" baseline hides future goal locations from the task-conditioned input. In theory, this effects AMAGO's ability to make exploration/exploitation trade-offs and take routes that maximize the chance of finding all $k$ locations, and does make a noticeable but small difference as the agent's strategy improves. However, $1,000$ timesteps may be too generous to highlight this trade-off.

**"Rewired" Action Space Adaptation.** Figure 22 shows the final performance of several methods with the randomly permuted action space feature enabled. Generalizing to new action dynamics requires a level of adaptation above standard Recurrent PPO, so our baselines shift to full meta-RL methods. While some multi-episodic methods can be modified to work in a zero-shot setting [111], we focus on three techniques that are more natural fits for this problem: variBAD [64], HyperX [111], and RL$^2$ [11, 10]. In an effort to use validated external implementations of these algorithms, we switch to the continuous action space version of MazeRunner. We choose the hyperparameters from the largest-scale experiment in the original codebase, with some tuning performed by solving easier

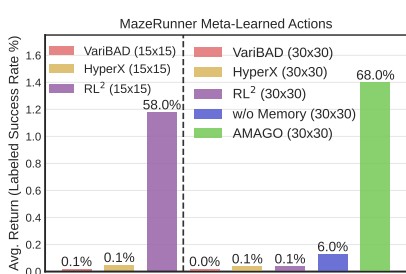

Figure 22: **MazeRunner with Random Dynamics.** The agent's action space is randomized in every environment.

versions of the problem. However, only RL$^2$ shows signs of life on $15 \times 15$ permuted actions (Fig. 22). $30 \times 30$ is too sparse for RL$^2$, but AMAGO's relabeling and long-sequence improvements allow us to nearly recover the metrics in Figure 8 while adapting to the randomized action space.

## C.5 CRAFTER

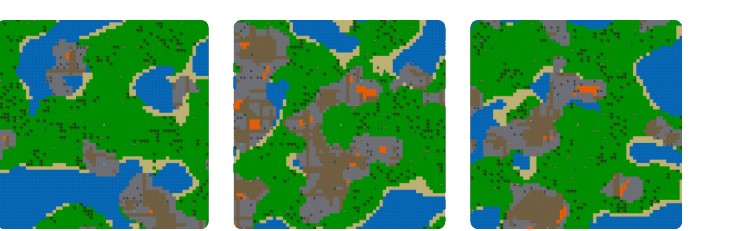 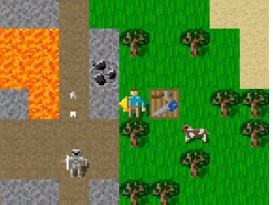

Figure 23: **Examples of Crafter Terrain Generation (Left).** Crafter generates new landscapes and locations of rare resources every episode (reproduced from [33]). **Crafter Observation (Right).** An example of the agent-centric view that forms observations.

Crafter [33] is a simplification of Minecraft with 2D graphics and a high framerate that facilitates research on building agents with the variety of skills necessary to survive and develop more advanced technology in a procedurally generated environment. The standard evaluation setting provides a lightly shaped dense reward that encourages survival and avoiding danger, while giving a sparse reward the first time the agent completes each of the 22 "achievements" in a new episode. This represents an *undirected* kind of multi-task learning, where the agent receives equal credit for completing any of its skills whenever the opportunity arises, and cannot be prompted with a specific objective. We create a *directed* version where the agent is only rewarded for completing the next step of an instruction we provide.

### C.5.1 ENVIRONMENT AND TASK DETAILS

**Environment Distribution** $p(e)$**:** Crafter generates a new world map that randomizes the locations of caves, lakes, and rare resources upon every reset. Three examples are shown in Figure 23.

**Goal Space** $\mathcal{G}$**:** We create goals from Crafter's 22 achievements, which include those listed in Table 1 as well as "collect sapling" (where all agents have a near $100\%$ success rate) and more difficult goals like "make iron sword" and "collect diamond" (where all agents have a near $0\%$ success rate). We add more complexity by creating goals for traveling to every $(x, y)$ location spaced 5 units apart, such as "travel to (5, 5)" and "travel to (60, 15)". An "Expanded" version of Crafter adds goals for placing blocks at every $(x, y)$ location. All goal strings are represented as three tokens – e.g., "<make> <stone> <sword>", "<travel> $< 50 > < 5 >$", "<collect> <coal> <PAD>" — which is meant to improve generalization across goals relative to one-hot identifiers.

**Observation Space:** Crafter observations are egocentric views of the agent and its surrounding area (Figure 23 right) along with health and inventory data. AMAGO supports pixel observations, but our experiments default to a version that maps the image (Figure 23 right) to texture IDs, which can be processed with an embedding rather than a CNN. This simplification shortens training times and lets us run many ablations, and is inspired by the NetHack Learning Environment [112] — another high-throughput generalization benchmark. However, we do evaluate the full AMAGO method on pixel observations with a CNN (Table 2), and the results are similar to the simplified version.

**Action Space:** Crafter has 17 discrete actions corresponding to player movement, tool making, and item placement.

**Instruction Distribution** $p(g \mid e)$**:** By default, our instructions are generated by randomly choosing an instruction length $k \sim U(1, 5)$ and then filling that length by weighted sampling (with replacement) from the goal space. We pick sample weights that ensured non-relabeling baselines could make some progress by down-weighting the hardest goals (like diamond collection). We defer the exact weights to our open-source code release. We also experiment with using the ground-truth Crafter achievement progression (see Crafter Figure 4 [33]) to generate instructions.

**Task Horizon** $H$**:** We enforce a maximum episode length of $H = 2,000$ timesteps. However, a typical episode is less than 500 timesteps, because the agents either succeed quickly or are defeated by an enemy.

### C.5.2 ADDITIONAL RESULTS AND ANALYSIS

Table 1 evaluates agents on single-goal instructions and lets us measure AMAGO's knowledge of the 22 main skills similar to existing Crafter results (Table 1 top section). We add a baseline version of our core agent trained with Crafter's default reward function to control for changes in state representation, sample limit, memory, and maximum episode length. Moving from the undirected reward function to sparse instruction-conditioning leads to a dramatic decline in performance without relabeling or long-term memory (Table 1 middle section). The lower section of Table 1 compares versions of our method with different instruction-generation and relabeling schemes, which will be explained below. AMAGO has learned to complete Crafter's achievement progression up to collecting iron despite the added difficulty that comes with being task-conditioned.

**Exploring with Instructions.**    The instruction format lets us discover new behavior by following sequences of steps from our own goal space, and turns exploration into a multi-task generalization problem. The process works in three steps. First, relabeling lets AMAGO master sequences of goals that are frequently achieved early in training. Second, we follow randomly generated instructions that happen to be made up of those easier goals, eventually reaching the frontier of what we have already discovered. Random exploration then has a realistic chance of finding new outcomes. We can verify the first two steps by ablating AMAGO's ability to follow multi-goal instructions during training (Table 1 "w/o Multi-Goal"), and observing a steep decline in performance on hard-exploration goals like tool making. In the third and final step, AMAGO learns to internalize the new discovery so that it can be achieved without the help of the instruction curriculum. We suspect this is closely tied with AMAGO's ability to observe the full instruction that led to the discovery, and test this by hiding the full task (Table 1 "w/o Planning"), which fails to discover hard goals despite following multi-goal instructions. The goal discovery process is supported by Figure 24, which measures the progress

| | Collect Wood | Collect Drink | Place Plant | Wake Up | Place Table | Wood Sword | Eat Cow | Defeat Zombie | Wood Pickaxe | Collect Stone | Place Furnace | Stone Pickaxe | Stone Sword | Collect Coal | Defeat Skeleton | Collect Iron |
|---|---|---|---|---|---|---|---|---|---|---|---|---|---|---|---|---|
| Rainbow (Undirected) [104] | 74.9 | 24.0 | 94.2 | 93.3 | 52.3 | 9.8 | 26.1 | 39.6 | 4.8 | 0.2 | 0.0 | 0.0 | 0.0 | 0.0 | 0.7 | 0.0 |
| DreamerV2 (Undirected)[86] | 92.7 | 80.0 | 84.4 | 92.8 | 85.7 | 40.2 | 17.1 | 53.1 | 59.6 | 42.7 | 1.8 | 0.2 | 0.3 | 14.7 | 2.6 | 0.0 |
| AMAGO (Undirected) | 99.9 | 93.3 | 99.9 | 95.8 | 99.8 | 99.1 | 81.1 | 91.3 | 99.4 | 97.5 | 93.6 | 86.3 | 92.3 | 69.5 | 53.5 | 0.0 |
| w/o Relabeling | 97.3 | 79.2 | **99.9** | 4.5 | 94.4 | 0.0 | 15.2 | 68.9 | 0.0 | 0.0 | 0.0 | 0.0 | 0.0 | 0.0 | 0.0 | 0.0 |
| w/o Memory | 99.8 | 91.3 | **99.9** | 91.5 | 98.9 | 98.4 | 62.2 | 85.7 | 99.2 | 39.3 | 0.0 | 0.0 | 0.0 | 0.0 | 0.0 | 0.0 |
| w/o Multi-Goal | 99.8 | 93.1 | **99.9** | 97.8 | 99.4 | 97.8 | 93.9 | 88.9 | 99.5 | 83.1 | 0.0 | 0.0 | 0.0 | 0.0 | 10.4 | 0.0 |
| w/o Planning | **99.9** | 98.1 | 98.7 | 93.6 | 95.7 | 93.9 | 36.0 | 82.0 | 98.6 | 0.0 | 0.0 | 0.0 | 0.0 | 0.0 | 0.0 | 0.0 |
| w/o Importance Sampling | **99.9** | 98.4 | 99.8 | 93.3 | 99.8 | **99.9** | **98.9** | 95.1 | **99.9** | 96.1 | 92.1 | 98.6 | 97.5 | 81.5 | 57.5 | 0.0 |
| AMAGO | **99.9** | **99.9** | 99.3 | **99.9** | 96.7 | 99.8 | 96.6 | 97.2 | 99.8 | **99.5** | 94.2 | 97.8 | 98.3 | **90.1** | **76.5** | 0.0 |
| AMAGO (w/ Tech Tree) | **99.9** | 98.3 | 99.8 | 96.1 | **99.9** | 99.8 | 98.5 | **97.3** | 99.8 | 97.8 | 97.0 | **99.0** | **99.5** | 84.0 | 58.4 | 0.0 |
| AMAGO (Expanded) | **99.9** | 97.4 | **99.9** | 94.3 | **99.9** | 99.8 | 97.9 | 96.7 | **99.9** | 98.6 | **97.7** | **99.0** | 99.0 | 84.8 | 40.4 | 0.0 |

Table 1: **Crafter Achievement Success Rates (%).** We compare *undirected* shaped reward agents (top section) with *directed* ablations of our agent (middle). AMAGO (bottom) recovers all the skills of the undirected reward function while being steerable/instruction-conditioned. The Rainbow/DreamerV2 vs. Undirected AMAGO comparison measures how skill coverage changes when using an increased sample limit, long-term memory, and simplified default observations.

of a single difficult goal relative to an easier multi-step instruction in which it is the final step. We note that Crafter's default reward function of giving $+1$ the first time each achievement occurs in an episode leads to this same exploration behavior. The best way for an agent to maximize the Crafter reward is to quickly enumerate every skill it has already mastered, and then start exploring for new rare possibilities. AMAGO's instruction format and relabeling scheme create a way to bring this behavior to a goal-conditioned setting.

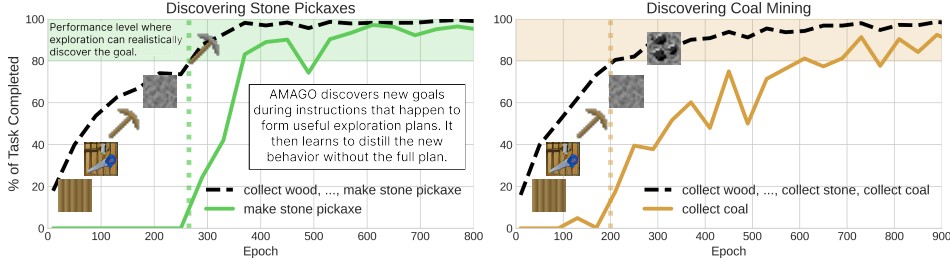

Figure 24: **Multi-Goal Learning and Exploration.** We compare AMAGO's performance on difficult single-goal instructions with another task that forms a useful exploration plan. We measure the average task progress, which is equivalent to return normalized by instruction length.

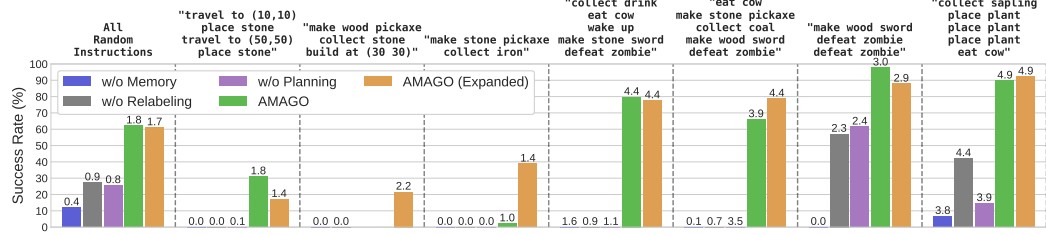

Figure 25: **Crafter Instruction Success Rates.** Bar labels are average returns. AMAGO generalizes well to user-prompted instructions in new Crafter environments.

Figure 25 expands on the results in Figure 9 to show more examples of user-selected multi-goal instructions in addition to "All Random Instructions", which corresponds to an expectation over the procedurally generated $p(g \mid e)$. Qualitatively, the agents show a clear understanding of instructions and typically fail by rushing to complete tasks during nighttime when they become surrounded by enemies.

**Impact of Goal Importance Sampling.** Learning curves in Crafter can have a sigmoid pattern where our agent plateaus for long periods of time before suddenly discovering a new skill that unlocks success in a higher percentage of tasks. Goal importance sampling shortens these plateaus by

prioritizing new skills during relabeling, and this leads to a significant increase in sample efficiency (Figure 26). Figure 27 demonstrates this goal prioritization on the kind of low-performance data AMAGO sees early in training. Unfortunately, all agents eventually reach the exploration barrier of consistently finding the iron and cannot complete Crafter's tech-tree. Below we discuss several preliminary attempts to address this problem.

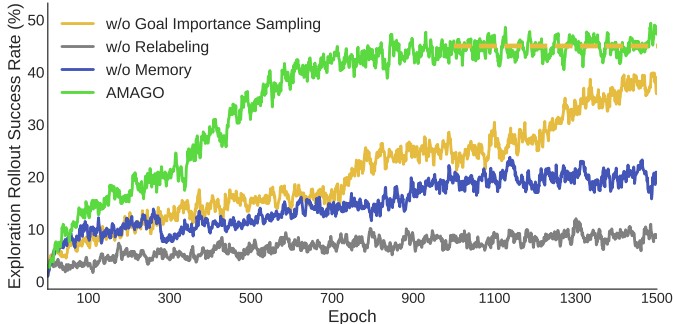

Figure 26: **Learning Curves in Crafter.** Goal importance sampling improves sample efficiency by shortening the "plateaus" between discoveries. However, both agents converge to the same exploration barrier (Table 1). "w/o Goal Importance Sampling" final performance is indicated by a dashed yellow line. This plot records *train-time* success rates and shows one random seed to highlight the sudden spikes in performance of individual runs.

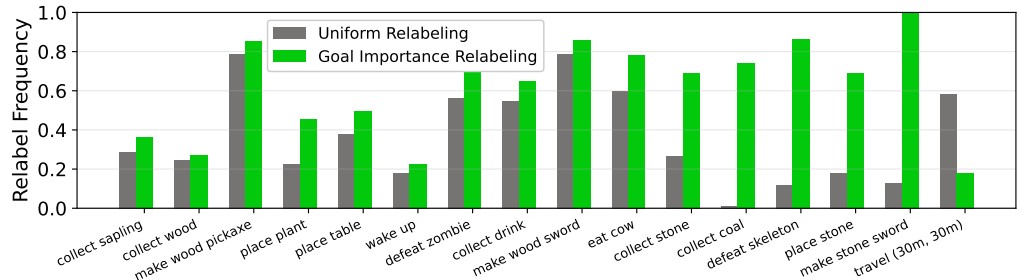

Figure 27: **Importance Sampling Alternative Goal Selection.** We use replay buffer data from a low-performance ablation ("w/o Planning") to demonstrate how our rarity-based relabeling scheme improves sample efficiency by prioritizing challenging goals. This plot is interpreted: "given that a particular goal is achieved in a trajectory that needs to be relabeled, what is the probability that it will be chosen to create an alternative instruction?" The rarity-based scheme exaggerates the frequency of challenging goals like "collect coal" and "make stone sword", and diminishes trivial goals like "travel (30m, 30m)." As the agent improves, goals that were once rare become more common and this effect is less extreme.

**Following "Tech-Tree" Instructions and "Expanded" Crafter.** While AMAGO learns many of Crafter's core skills, it still converges without mastering all of the goal space. Our results lead us to believe there are two main components driving AMAGO's exploration, and it is not clear which is the main bottleneck. First is the instruction distribution $p(g \mid e)$ that forms exploration plans for new goals. The exploration effect relies on generating useful instructions that can lead to new discoveries, and could be improved if the instructions we provide are more likely to be helpful. We can use ground-truth knowledge of Crafter's skill progression to bias $p(g \mid e)$ towards instructions that form useful exploration plans. In this "Tech-Tree" version, half of the instructions are generated by sampling sub-sequences of Crafter's progression up until the most advanced goal we have previously achieved. Unfortunately, this does not lead to the discovery of new skills (Table 1 "w/ Tech-Tree"). The second opportunity is to expand the goal space that forms exploration curricula. "Expanded" Crafter experiments with this idea and creates tech-tree instructions with added goals for being nearby

rare resources like iron and diamond. This does not fully solve the problem, but shows some signs of improvement, as multi-goal prompts like "make stone pickaxe, collect iron" now find iron nearly 40% of the time (Figure 25). Finding better ways to improve instruction generation (preferably without the domain knowledge used here) and extend the goal space by learning in more complex multi-task domains are exciting directions for future work.

**Planning and Long-Horizon Tasks.**  AMAGO's ability to see all $k$ steps of its instruction is meant to allow value learning to form long-term plans. If we only provide the current step, our instruction is effectively handling all the agent's planning. However, if we provide the full instruction, AMAGO is free to explore its environment with the future in mind and take actions that are not directly related to completing the task but will better prepare it for later events. Figure 28 measures the success rate at increasing instruction lengths. We can use the fact that our default instruction distribution samples each step independently to get a rough upper bound on performance. This is not a fair expectation because each additional goal extends the length of the episode and leaves more time for starvation or dangerous enemies to cause failure. However, AMAGO holds to this line quite well despite solving complex goals that take hundreds of timesteps to complete. We would ideally see the "w/o Planning" ablation fall well below its

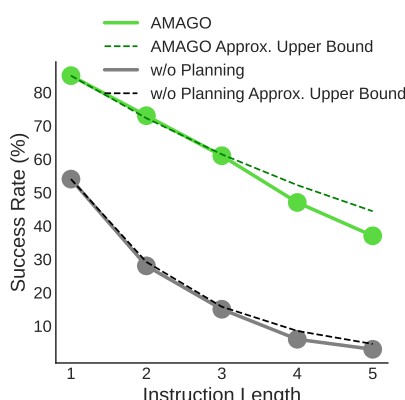

Figure 28: **Impact of AMAGO's (Implicit) Planning on Crafter Instructions.**

upper bound because it would be less careful about resource management. However, it learns to solve so many fewer complex goals to begin with that looking at long instructions that succeed reduces us to a sample of mostly trivial tasks that would not require much planning at all.

**Instruction-Specific Behavior.**  One concern is that because Crafter only has 22 different core skills, our agents could learn an uninteresting strategy of ignoring their instruction and cycling through every behavior they have learned to accomplish. This is one of the reasons we add travel goals, which brings the total goal count high enough where this strategy would be very difficult to learn and execute. "Expanded" Crafter also includes goals for placing stone blocks at each $(x, y)$ location. Some of the manually-generated instructions in Figure 25 were specifically chosen to be unrelated to the natural progression of Crafter's achievement system. For example, "travel to (10, 10), place stone, travel to (50, 50), place stone" requires the agent to traverse nearly the full length of its world and would not involve advanced resource-gathering. This task is difficult because long-distance travel attracts dangerous enemies, but even when AMAGO fails it does so with a clear understanding of what was being asked and does not waste time on unrelated objectives.

**Relabeling Corrects for Missing Instructions.**  An advantage of hindsight instruction relabeling is that the train-time $p(g \mid e)$ it generates includes every goal that is accomplished, rather than just those we assign during rollouts. This lets us learn about instructions that could not be generated by the domain's natural instruction distribution. We create a simple demonstration of this by removing the "wake up" goal from our instruction generation. Because this goal will occur unintentionally, AMAGO learns to complete it when requested 99.9% of the time (Table 1). However, the "w/o Relabeling" ablation has never seen these goal tokens and its 4.5% success rate represents the episodes where it is achieved by chance due to a stochastic policy.

| | Collect Sapling | Collect Wood | Collect Drink | Place Plant | Wake Up | Place Table | Place Furnace | Eat Cow | Defeat Zombie | Wood Pickaxe | Wood Sword | Collect Stone | Stone Pickaxe | Stone Sword | Collect Coal | Defeat Skeleton | Collect Iron | All Random Instructions |
|---|---|---|---|---|---|---|---|---|---|---|---|---|---|---|---|---|---|---|
| Textures | 99.9 | 99.9 | 99.9 | 99.3 | 99.9 | 99.9 | 94.5 | 96.2 | 96.6 | 99.9 | 99.9 | 99.5 | 97.8 | 98.3 | 90.1 | 76.5 | 0.0 | 62.1 |
| Pixels | 99.9 | 99.9 | 99.4 | 99.9 | 93.7 | 99.5 | 94.0 | 99.3 | 96.7 | 99.9 | 99.9 | 98.3 | 98.8 | 97.7 | 89.8 | 80.0 | 0.0 | 53.9 |

Table 2: **AMAGO Crafter Success Rates (%) from Pixels.**

**Learning from Pixels.**  Our main experiments map Crafter image observations to discrete block textures. This is primarily a compute-saving measure that lets us study more ablations. However, we do evaluate the full AMAGO method on pixel observations, with a summary of key results in Table

2. Learning from pixels performs similarly to texture observations on all single-goal instructions. However, there is a roughly $9\%$ drop in performance on the full range of instructions $p(g \mid e)$.

# D  POLICY ARCHITECTURE AND TRAINING DETAILS

**AMAGO Hyperparameter Information.**  Network architecture details for our main experimental domains are provided in Table 3. Table 4 lists the hyperparameters for our RL training process. Many of AMAGO's details are designed to reduce hyperparameter sensitivity, and this allows us to use a consistent configuration across most experiments.

| | POPGym | Dark Key-To-Door | Wind | Passive T-Maze | Meta-World | Package Delivery | MazeRunner | Crafter |
|---|---|---|---|---|---|---|---|---|
| **Transformer** | | | | | | | | |
| Model Dim. | 256 | 256 | 128 | 128 | 256 | 128 | 128 | 384 |
| FF Dim. | 1024 | 1024 | 512 | 512 | 1024 | 512 | 512 | 1536 |
| Heads | 8 | 8 | 8 | 8 | 8 | 8 | 8 | 12 |
| Layers | 3 | 3 | 2 | 2 | 3 | 3 | 3 | 6 |
| **Other Networks** | | | | | | | | |
| Actor MLP | (256, 256) | (256, 256) | (256, 256) | (256, 256) | (256, 256) | (256, 256) | (256, 256) | (400, 400) |
| Critic MLP | (256, 256) | (256, 256) | (256, 256) | (256, 256) | (256, 256) | (256, 256) | (256, 256) | (400, 400) |
| Goal Emb. | N/A | N/A | N/A | N/A | N/A | FF (64, 32) | FF (64, 32) | RNN (Hid. Dim 96) $\rightarrow$ 64 |
| Timestep Encoder | (512, 512, 200) | (128, 128, 64) | (128, 128, 64) | (128, 128, 128) | (256, 256, 200) | (128, 128, 128) | (128, 128, 128) | Embedding $\rightarrow$ MLP (384, 384) |

Table 3: **AMAGO Network Architecture Details.**

| | Package Delivery | MazeRunner | Crafter | Other Domains (POPGym) |
|---|---|---|---|---|
| Critics | | | 4 | |
| Critics in Clipped TD Target | | | 2 | |
| Context Length $l$ | | | $H$ | |
| Actor Loss Weight | | | 1 | |
| Filtered BC Loss Weight | | | .1 | |
| Value Loss Weight | | | 10 | |
| Multi-Gamma $\gamma$ Values (Discrete) | | .7, .9, .93, .95, .98, .99, .992, .994, .995, .997, .998, .999, .9991, .9992, . . . , .9995 | | |
| Multi-Gamma $\gamma$ Values (Continuous) | | .9, .95, .99, .993, .996, .999 | | |
| Target Update $\tau$ | | | .003 | |
| Gradient Clip (Norm) | | | 1 | |
| Learning Rate | 3e-4 | 3e-4 | 1e-4 | 1e-4 |
| L2 Penalty | 1e-4 | 1e-4 | 1e-4 | 1e-3 |
| Batch Size (in Full Trajectories) | 24 | 24 | 18 | 24 |
| Max Buffer Size (in Full Trajectories) | 15,000 | 20,000 | 20,000 | 20,000 (80,000) |
| Gradient Updates Per Epoch | 1500 | 1000 | 1500 | 1000 |
| Parallel Actors | 24 | 24 | 8 | 12 (24) |
| Timesteps Per Actor Per Epoch | $H$ | $H$ | $H$ | $H$ (1000) |
| Epochs | 600 | 600 | 2000 | (625) |
| Exploration Max $\epsilon$ at Ep. Start | | | $1. \rightarrow .05$ | |
| Exploration Max $\epsilon$ at Ep. End | | | $.8 \rightarrow .01$ | |
| Exploration Annealed (Timesteps - Per Actor) | | | $1,000,000$ | |

Table 4: **AMAGO Training Hyperparameters.**

**Compute Requirements.**  Each AMAGO agent is trained on one A5000 GPU. Results are reported as the average of at least three random seeds unless otherwise noted. Learning curves default to displaying the mean and standard deviation of trials. The POPGym (Appendix C.1) and Wind (Figure 5) learning curves show the maximum and minimum values achieved by any seed. Wall-clock training times vary significantly across experiments and were improved over the course of this work. For reference, POPGym training runs take approximately 8 hours to complete. AMAGO alternates between data collection and learning updates for consistent comparisons across baselines. However, these steps could be done in parallel or asynchronously if wall-clock speed was critical.

**Sample Efficiency.**  The AMAGO training loop alternates between collecting trajectory rollouts from parallel actors and performing learning updates. Sample efficiency in off-policy RL is primarily determined by the update-to-data ratio between these two stages [79, 74]. It has been shown that common defaults are often too conservative and that sample efficiency can be greatly improved by

increasing the update-to-data ratio [113, 80]. AMAGO's use of Transformers resets this hyperparameter landscape. Due to compute constraints and the large number of technical details we are already evaluating, our experiments make little effort to optimize sample efficiency. It would be surprising if the current results represent the best trade-off between sample efficiency and performance.

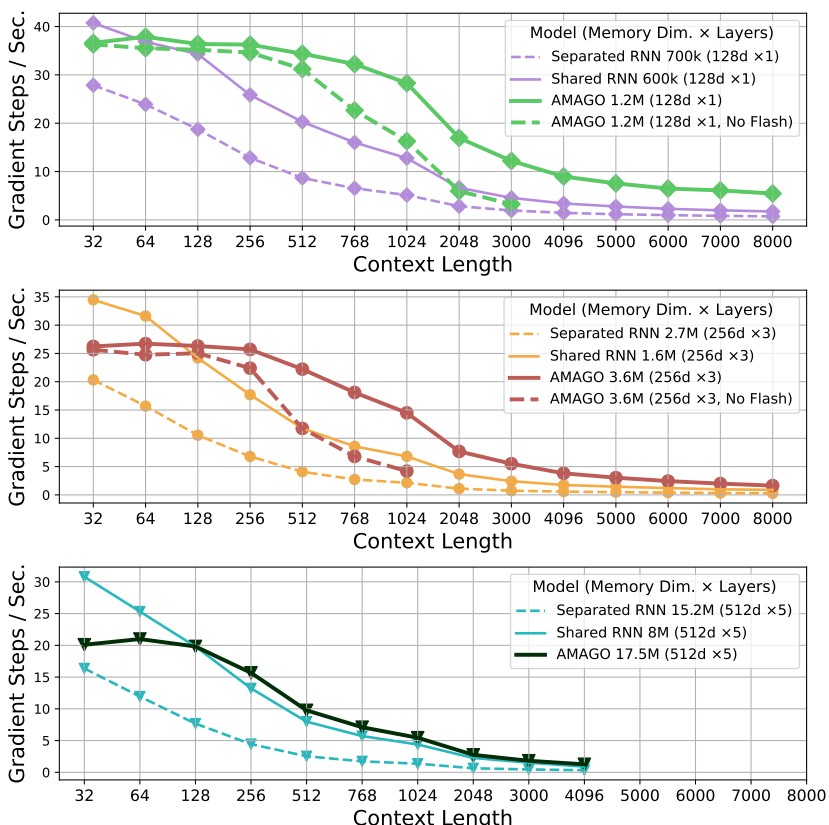

Figure 29: **Training Throughput: AMAGO vs. Off-Policy RNNs.** We compare training speed in terms of gradient updates per second at various context lengths with a batch size of 24 sequences on a single NVIDIA A5000 GPU. These experiments use a HalfCheetah locomotion environment [97] and the codebase from [22] to benchmark the RNN agents. We modify AMAGO's hyperparameters to create a fair comparison, though AMAGO is loading data from disk while the RNNs are loading from RAM. "No Flash" baselines remove Flash Attention [108].

