# OpenReview forum: "AMAGO: Scalable In-Context Reinforcement Learning for Adaptive Agents"
_ICLR.cc/2024/Conference — ICLR 2024 spotlight_

### Official Review · Reviewer_WXv6 · 2023-10-14

**Soundness:** 3 good
**Presentation:** 3 good
**Contribution:** 4 excellent
**Rating:** 8
**Confidence:** 4

**Summary:**

This paper proposes an RL agent for POMDPs. The proposed agent uses a transformer architecture to handle long context lengths. The agent is trained using a new off-policy RL update, which combines elements of previously proposed actor-critic updates. The update is designed to require relatively little hyperparameter tuning and reduce the amount of computation in the sequence model compared to relevant baselines. The proposed agent is tested on meta-RL, goal-conditioned RL, and other POMDP benchmarks. It achieves strong performance on the recently proposed Crafter environment.

**Strengths:**

## Contribution
- Training transformers for POMDPs is a popular subject in research. Good results in this area have the potential for significant impact on RL research and hopefully beyond, as POMDPs are a very general problem class. Based on the strong results in the Crafter environment, this paper seems to push the state-of-the-art on that front, at least in the realm of open source research, for which it is to be commended.
- The paper includes a suite of experiments in a variety of environments, which probe the different challenges arising from in partially observable settings. Benchmarking the algorithm on a diverse set of environments is good and helps establish the robustness of the proposed method.

## Presentation
- The introduction, background, and related work sections are well written.

**Weaknesses:**

## Presentation
- This paper is perhaps trying to do too much in the limited space to the detriment of the clarity of its most important contributions. Given that it is in the method name, goal-conditioned RL seems to be an important part of the contribution. However, this paper would be much easier to understand if it focused on the meta-RL part, where it already has strong results, and upon which its success in goal-conditioned RL builds.
- The proposed agent uses a new combination of existing actor critic losses to construct its update function, yet the main paper does not contain almost any detail about how the objective is motivated or what the objective even is. Contrast this, for example, to the DreamerV3 paper (Mastering Diverse Domains through World Models, Hafner et al. 2023), which tackles a similar problem setting, where the majority of the paper is describing the method and motivating the design choices. Even after reading the relevant parts of the appendix, it is not clear to me how exactly is the critic trained.
- The architecture design is described as being based on "clear pattern of network activations across hundreds of trials" but this pattern is not described anywhere.

## Soundness
- The proposed method is quite complicated including both architecture and objective function innovations. Some of the ideas are not super well motivated or ablated.
    - Popart is proposed to help keep the different losses on the same scale across different environments, but I didn't find any attempt to validate that it is indeed doing so or helping in any other way.
    - A complicated exploration schedule is proposed and used, but at the same time the paper says it is not essential.

**Questions:**

- The paper explicitly claims state-of-the-art results on PoPGym but refrains from doing so for Crafter, where it also seems to be beating the previous state-of-the-art (Dreamerv3). Why?
- Have I understood it correctly that the "dense" Crafter is the vanilla Crafter setting?
- Why not include results for Dreamerv3 for Crafter?
- What does "relegate to an unstable baseline" in the introduction mean?
- What is the actor objective used by the method?
- What is the critic objective used by the method?
- What is "the clear pattern of network activations across hundreds of trials"?
- This is very subjective so I'm not listing this as a weakness, but I find the "in-context learning" label to make it less clear what the paper is about. On an abstract level, the method is a variant of well-known black-box meta-RL methods like RL2. Black-box meta-RL methods have variously been grouped under other names like context-based meta-RL, memory-based meta-RL, zero-shot meta-RL, etc. Given that this is a long ongoing research direction in the RL community, with an already confusing diversity of labels, it's not clear to me that emphasizing a new label makes this easier to understand. Granted that for non-RL audiences "in-context learning" may be more intuitive than "black-box meta-RL". ADA uses the term similarly, but when they get to the technical details, they switch to talking about black-box meta-RL.

**Details Of Ethics Concerns:**

No concerns

---

> ### Author Response · Authors · 2023-11-18
> **Reply to Reviewer WXv6 (Part 1)**
>
> Thanks for your review, especially of the Crafter appendix section. Many of your questions focus on technical details that require some explanation. We will try to keep this reply brief but it will need to be split into multiple parts.
>
> > "goal-conditioned RL seems to be an important part of the contribution. However, this paper would be much easier to understand if it focused on the meta-RL part"
>
> With this suggestion in mind our revision has prioritized the first experiment section (meta-RL and long-term memory) when dealing with space constraints. We have moved one of the goal-conditioned section’s figures to the Appendix.
>
> > "the main paper does not contain almost any detail about how the objective is motivated or what the objective even is. Contrast this, for example, to the DreamerV3 paper …, where the majority of the paper is describing the method … "
>
> We have revised the Method section to help clarify the motivation of our learning update. More minor RL engineering details that are not discussed in the main text or Appendix A are much easier explained in code/documentation, and we assure you that our agent is fully open-source. To be fair, DreamerV3 is building upon a series of implementation-focused papers and it is quite the outlier in terms of how much low-level detail is explained in the text.
>
> > "The architecture design is described as being based on "clear pattern of network activations across hundreds of trials" but this pattern is not described anywhere."
>
> Yes, this was too vague of an explanation, and it has now been updated in the main text and expanded in the Appendix alongside a new figure (Fig. 14).
>
> > "Popart is proposed to help keep the different losses on the same scale across different environments, but I didn't find any attempt to validate that it is indeed doing so or helping in any other way."
>
> Appendix A.2 Equation 1 shows the weighted combination of three terms used to create AMAGO's shared loss. The actor and critic terms scale with the learned $Q$ value, but the filtered BC term does not. This means that the relative weight of the three terms is based on the scale of rewards in any given environment and would require constant tuning. Popart creates a simple solution by normalizing $Q$ during loss computation. The loss scales then converge to predictable values and we validate this by using the exact same weights in every experiment. If you think it would be helpful we can demonstrate this by directly scaling rewards in the same environment (this is how the idea was first tested). Popart saves time, compute, and frustration when evaluating many environments but does not meaningfully change the method if we only care about a specific environment. We are *not* using Popart to independently normalize the returns of multiple tasks simultaneously (like how it is used in Atari) because the size of our task distribution is assumed to be unknown and large.
>
> > "A complicated exploration schedule is proposed and used, but at the same time the paper says it is not essential."
>
> The correct amount of exploration noise does not usually depend on the current timestep. However, in meta-RL, each rollout creates its own exploration-exploitation tradeoff as we try to identify the current environment. We feel the annealed schedule is a stronger default here, especially because it can be configured to be the same as regular $\epsilon$-greedy if necessary. However, tuning this schedule on each environment goes against our goal of creating a baseline that is easy to use. We did not prioritize this and leave it as an Appendix detail. We pick a conservative default schedule randomized over parallel actors so that performance in tasks with easy exploration can be mostly unaffected. This schedule would likely be more important if our experiments were more concerned with sample efficiency than stability and final performance.

---

> > ### Author Response · Authors · 2023-11-18
> > **Reply to Reviewer WXv6 (Part 2 - Crafter)**
> >
> > > "The paper explicitly claims state-of-the-art results on PoPGym but refrains from doing so for Crafter …. Why?"
> >
> > The difference is that we match the evaluation protocol of existing POPGym baselines as closely as possible but are not doing the same for Crafter. Instead, we are using the Crafter world generation and achievement system to create our own evaluation setting for goal-conditioned agents in an open-world domain.
> >
> > > "Have I understood it correctly that the "dense" Crafter is the vanilla Crafter setting?"
> >
> > Yes you are right, and the term “dense” in Appendix Table 1 comes from the Crafter rewards having some shaping aside from the binary achievements. However, this is probably a poor word choice on our part, as these rewards are still sparse by most standards. The text uses the term “undirected,” and we have now updated the table to match. The most important difference is that Crafter usually rewards the agent for completing any achievement, while our version rewards completing a specific task that is provided.
> >
> > > "Why not include results for Dreamerv3 for Crafter?"
> >
> > The Rainbow/DreamerV2 results in Appendix Table 1 for the regular (“undirected”) version of Crafter are mainly included to determine how our other evaluation changes adjust the upper bound of performance before adding the extra challenge of goal-conditioning. This is referenced in the paragraph above the table, but we have made it more clear with a longer caption in the revision.
> >
> > We have not been able to find the raw scores for DreamerV3 (the paper lists them on a log scale bar chart). Given what we are trying to accomplish, it might be more useful to only list the Rainbow scores because it is a more comparable learning update to AMAGO.

---

> ### Author Response · Authors · 2023-11-18
> **Reply to Reviewer WXv6 (Final Part)**
>
> > "What does "relegate to an unstable baseline" in the introduction mean?"
>
> We mean that the black-box/in-context approach was an early idea but that it became a weak baseline in many of the followup meta-RL papers in the years after RL^2.
>
> > "What is the actor objective used by the method?"
>
> It is a mixture of the standard $-Q(s, \pi(s))$ taking the min over 2 critics from an ensemble like TD3/REDQ and filtered behavior cloning. The behavior cloning term is motivated by the same idea as multi-gamma training, where we are trying to prevent learning from becoming unstable or uninformative when the Q-surface is flat. Hopefully the method section writing changes have helped clarify this.
>
> > "What is the critic objective used by the method?"
>
> It is standard one-step temporal difference error implemented with an ensemble of Q-networks to reduce overestimation like in REDQ. We use REDQ to reduce the risk of picking an update-to-data ratio that is too high, which can happen early in training because we are replaying entire rollouts in every update. It is not uncommon for the agent to perform gradient updates on the entire replay buffer several times between rollouts. There are many open questions here (Appendix D - final paragraph).
>
> > "What is "the clear pattern of network activations across hundreds of trials"?"
>
> Hopefully we have addressed this question in Part 1.
>
> > "I find the "in-context learning" label to make it less clear what the paper is about. On an abstract level, the method is a variant of well-known black-box meta-RL methods like RL2. Black-box meta-RL methods have variously been grouped under other names like context-based meta-RL, memory-based meta-RL, zero-shot meta-RL, etc. …, it's not clear to me that emphasizing a new label makes this easier to understand. Granted that for non-RL audiences "in-context learning" may be more intuitive than "black-box meta-RL". ADA uses the term similarly, but when they get to the technical details, they switch to talking about black-box meta-RL."
>
> We completely agree that the terminology is confusing and thought a lot about how we should handle this. There are several names for the method and even several names for the environment type and problem statement. Ultimately, a key advantage of this kind of black-box method is that it works for meta-RL *and* zero-shot generalization *and* long-term memory while other meta-RL methods may not. This term feels too restrictive for our work because our experiments emphasize that flexibility more than most meta-RL papers. We decided to call the environments CMDPs for a similar reason (that term is more common in generalization than meta-RL). So, if we are not going to use "meta-RL",  it seems reasonable to use "in-context RL" because "in-context learning" has become a common term for Transformers outside of RL as you note. And at least we are not adding to the problem by creating yet another name because AdA uses the same term. However, we do make a point of using a more technical term in the related work section for those with a meta-RL background: "Meta-RL is a crowded field with a complex taxonomy; this paper uses the informal term "in-context RL" to refer to a specific subset of implicit context-based methods that treat meta-learning, zero-shot generalization, and partial observability as a single problem" followed by many meta-RL references.

---

> > ### Author Response · Authors · 2023-11-19
> > **Status on Revisions to Appendix A**
> >
> > We have decided it would be best to add more of the technical details discussed in your review and our reply to Appendix A, including a new figure demonstrating the varying scale of AMAGO's loss terms and how PopArt solves this problem. A few experiments are still wrapping up, but since we are nearing the end of the discussion period we would like to give advance notice that we will be updating the revision with these changes tomorrow (Monday 11/20).

---

> > > ### Author Response · Authors · 2023-11-20
> > > **Changes to Appendix A.2**
> > >
> > > We have rewritten Appendix A.2 (the RL update section) to be significantly more detailed and the pdf has been updated. We hope that this version answers most of your questions about the learning update and related RL engineering details.

---

> > > > ### Comment · Reviewer_WXv6 · 2023-11-21
> > > >
> > > > Thanks for the detailed response. The revised appendix is clear and answers most of my questions. The rest were answered in your response. I raise my score.

---

### Official Review · Reviewer_LVWP · 2023-10-27

**Soundness:** 3 good
**Presentation:** 4 excellent
**Contribution:** 2 fair
**Rating:** 8
**Confidence:** 3

**Summary:**

The authors apply a transformer to in-context reinforcement learning, effectively producing a smaller-scale and open-source variant of Adaptive Agents.

Observations, actions, rewards, episodic resets, goals, and the timestep are encoded into a fixed-size token for a transformer. The authors feed a long sequence consisting of multiple episodes to the transformer to produce a latent representation. Actor and critic heads process this representation into actions and values respectively.

The authors propose a series of tricks to help stabilize transformers in RL:
- Multiple discount factors
- Shared transformer for actor/critic
- Hindsight relabeling

I believe this is a generally useful paper, even if the novelty and theory is a bit lacking.

**Strengths:**

- The paper is well written and easy to follow
- The authors set a new record on a standard benchmark, and in general provide a large number of experiments across many environments
- There are a large number of ablation studies
- This is effectively a more useful version of the AdA paper that does not rely on closed-source benchmarks and does not require policy distilation or corporation-scale resources, making it much more useful to the academic community

**Weaknesses:**

- The high-level concept is not novel -- applying transformers to POMDPs or for in-context RL has been heavily studied
- This method still requires server-grade GPUs to run, limiting this approach to well-off labs or industry
- For the POPGym benchmark, they are comparing their off-policy method to an on-policy method given the same number of sampled timesteps. This is not a fair comparison, but they do label the baselines as on-policy.
- They do not list any shortcomings

**Questions:**

- I would like the authors to list any shortcomings of their approach, especially because this paper is mostly empirical

---

> ### Author Response · Authors · 2023-11-17
> **Reply to Reviewer LVWP**
>
> Thank you for your review! We address your questions and concerns below.
>
> > "The high-level concept is not novel -- applying transformers to POMDPs or for in-context RL has been heavily studied"
>
> Yes, while this concept is not new, our work is much more about the details necessary to get it to work over long sequences with Transformers in an off-policy setting.
>
> > "This method still requires server-grade GPUs to run, limiting this approach to well-off labs or industry"
>
> AMAGO is more GPU-dependent than many single-task baselines (for example), but we will note that every experiment is performed on a single GPU. We inherit the same GPU memory bottleneck as most other work on Transformers, but there are many hyperparameter choices that can help address this problem (batch size, context length, model size, …).
>
> > "For the POPGym benchmark, they are comparing their off-policy method to an on-policy method given the same number of sampled timesteps. This is not a fair comparison, but they do label the baselines as on-policy."
>
> We agree that off-policy methods have an advantage in any sample-limited benchmark, but this can be considered a strength of the method. The sample limit in POPGym is a bit arbitrary as it was established by on-policy baselines that did not solve many of the tasks. The [POPGym paper](https://openreview.net/pdf?id=chDrutUTs0K) Figures 10-14 show that learning curves on many environments have not improved or converged long before the 15M sample limit. One positive sign of our work is that performance is still increasing on many of the harder environments (Appendix C.1.). This creates a convenient opportunity for research because now that we have a way to reach strong performance at convergence we can start experimenting with how to reach those levels faster.
>
> > "I would like the authors to list any shortcomings of their approach"
>
> One shortcoming of the current method is that the goal relabeling scheme only applies to sparse/binary rewards for goal completion. However, we believe this could be expanded to arbitrary reward functions. Many of the shortcomings of (value-based) RL still apply. For example, long-term credit assignment can still be challenging when the discount factor is not the only underlying problem. We are also beginning to saturate the memory lengths of standard benchmarks, as the sequence lengths used in our experiments are often longer than successful solutions to the same/similar problems in prior work. A key motivation of AMAGO was to create an open-source in-context learner that can help create new benchmarks in this area.

---

> > ### Comment · Reviewer_LVWP · 2023-11-23
> >
> > Thank you for the response.

---

### Official Review · Reviewer_t9aE · 2023-10-30

**Soundness:** 3 good
**Presentation:** 2 fair
**Contribution:** 4 excellent
**Rating:** 8
**Confidence:** 3

**Summary:**

The authors present AMAGO, a method that successfully trains long-range Transformers over entire rollouts using off-policy RL. The authors also combine AMAGO with a hindsight relabeling scheme. The authors demonstrate strong performance on a variety of meta-RL and long-horizon tasks.

**Strengths:**

- AMAGO improves over baselines on long-horizon tasks while still performing well on simpler tasks.
- The authors perform a series of ablations in the Appendix to isolate the effect of each design choice.
- The authors test AMAGO on a very thorough set of experiments and show good results.

**Weaknesses:**

- It is a bit hard to keep track of what exactly the different components of the proposed method are. I think it would greatly improve the paper's clarity if the authors could include a table outlining exactly what the contribution is (e.g. Transformer architecture change, multi-gamma update, relabeling, etc.). Generally, I think the main improvement the authors can make on this paper is clarity.
- Similarly, an algorithm box for the relabeling scheme would be nice for clarity. I know the authors say they defer details of the relabeling to the open source code release, but I think it would be good to have it in the appendix at least.
- The proposed method seems quite complex. Generally, simpler methods should be preferred. However, this is mitigated by the good performance and the ablation studies.

**Questions:**

- The authors write that "AMAGO can select the action corresponding to any $\gamma$ at test time". How exactly is this done?
- Have the authors tried experimenting with a smaller set of multi-gamma values for the multi-gamma update rather than using all the values in Table 4? I'm curious if/how much performance would suffer.

---

> ### Author Response · Authors · 2023-11-17
> **Reply to Reviewer t9aE**
>
> Thank you for your review! We address your questions and concerns below and would be happy to continue the discussion.
>
> > "It is a bit hard to keep track of what exactly the different components of the proposed method are…. Generally, I think the main improvement the authors can make on this paper is clarity."
>
> We have updated the Transformer architecture change and multi-gamma update sections of the Method section, which we hope will improve their clarity.
>
> > "Similarly, an algorithm box for the relabeling scheme would be nice for clarity."
>
> We have taken your suggestion and added an algorithm box in Appendix B, along with an expanded discussion of the method. The relabeling scheme extends Hindsight Experience Replay (HER)  to multiple steps where we select sub-sequences of alternative goals. The main details are 1) how hindsight goals are selected and 2) how they are merged into the original sequence of real goals. The multi-step version gives more room for customization than HER; for example, it creates a spectrum between relabeling every trajectory to be a complete success and not relabeling at all.
>
> > "The authors write that "AMAGO can select the action corresponding to any $\gamma$  at test time". How exactly is this done?"
>
> During training, our actor and critics networks are outputting values for an array of different $\gamma$ values in parallel. At test-time, we can select the index of the actor’s outputs that corresponds to any value of $\gamma$. This may be simpler than it sounds because computing RL losses over long causal sequences involves lots of parallel computation (over multiple timesteps, multiple critics, etc.), and the multi-gamma method adds one more axis to this sequence format.
>
> > "Have the authors tried experimenting with a smaller set of multi-gamma values for the multi-gamma update rather than using all the values in Table 4?"
>
> The multi-gamma technique is efficient, so once we identify the optimization of a single $\gamma$ value as an issue, it is convenient to add many more values. The most important requirement for AMAGO’s technical details is that learning remains stable even when we default to somewhat extreme settings for its hyperparameters – this makes hyperparameter selection less risky and is what enables us to evaluate on a wide range of domains. It is quite possible that the Table 4 settings are well into the diminishing rate of returns of this technique for some environments. We tried running this ablation for this response, but in the interest of time, we picked an environment that was too simple to create a clear demonstration. We will continue these experiments and update the paper with more information.

---

> > ### Comment · Reviewer_t9aE · 2023-11-23
> >
> > Thanks for the clarifications and the changes! I find that clarity, which was my main concern, has been improved, so I am raising my score.

---

### Official Review · Reviewer_KbVB · 2023-10-30

**Soundness:** 3 good
**Presentation:** 3 good
**Contribution:** 3 good
**Rating:** 6
**Confidence:** 4

**Summary:**

This paper presents AMAGO, which is designed to tackle challenges related to generalization, long-term memory, and meta-learning. It achieves this with the use of sequence models. The paper shows that AMAGO can successfully train long-sequence Transformers, able to perform tasks over entire rollouts, while operating with end-to-end RL.

The primary contributions are twofold. Firstly, it revises the off-policy in-context approach and deploys sequence models like Transformers to replace the recurrent policies. This change facilitates the tackling of constraints related to memory capacity, model size, and the planning horizon in agents. Secondly, AMAGO extends in-context learning to goal-conditioned problems, making it capable of handling more sophisticated exploration tasks. The authors validate AMAGO's performance through empirical evaluations, demonstrating capabilities in large-scale challenges requiring long-term memory and adaptation.

**Strengths:**

- By redesigning the off-policy in-context approach and using sequence models, AMAGO overcomes previous bottlenecks in memory capacity, planning horizon, and model size.
- The introduction of a hindsight relabeling scheme broadens the applicability and scalability of this approach to open-world environments.
- The evaluations illustrate capabilities of AMAGO in diverse large-scale, heavy-memory and meta-learning RL challenges.
- The structure and presentation of the paper are clear and well-organized.

**Weaknesses:**

- The AMAGO architecture's complexity, particularly due to the use of transformers, can be computationally intense. This may limit its efficiency in resource-limited situations or possibly make it less suitable for complicated applications.
- Although AMAGO has made improvements in performance, it still has a success rate of 0 in collecting some key resources (e.g. iron), so AMAGO still faces limitations in exploration challenges.

**Questions:**

- Considering AMAGO's ability to work with multi-goals, how well does AMAGO prioritize or sequence goals when given multiple objectives? Is there a mechanism to manage potential conflicts between goals?
- Are there any experimental results that can support the claim that AMAGO does not require much tuning?
- The author mentioned replacing ReLU/GeLU with Leaky ReLU, adding LayerNorms, and modifying linear layers. It may be better to explain the necessity and rationality behind each modification through a more thorough ablation study.

---

> ### Author Response · Authors · 2023-11-17
> **Reply to Reviewer KbVB**
>
> Thank you for your review! We address your questions and concerns below. Please let us know if more clarification is needed.
>
> > "…the use of transformers, can be computationally intense. This may limit its efficiency in resource-limited situations…"
>
> Transformers are more expensive than common RL architectures, but this approach to generalization does require some kind of memory, so “expensive” becomes a relative term. We show in Figure 6 (previously Fig. 4)  that Transformers can be an efficient choice vs RNNs in the common RL setting of small model sizes and long sequence lengths.
>
> > "Although AMAGO has made improvements in performance, it still has a success rate of 0 in collecting some key resources (e.g. iron), so AMAGO still faces limitations in exploration challenges."
>
> We do make some progress on iron collection in Appendix C.5 (Figure 23 and the paragraph titled "Following Tech-Tree Instructions and Expanded Crafter" on page 30) although this may be the experiment you are already referring to. Our work tries to focus more on the memory/generalization problem of RL with sequence models than pure exploration. We agree that more improvements in exploration are needed. However, successful adaptive agents create new ways to drive exploration by transferring knowledge between similar tasks.  The Crafter experiments also highlight helpful advantages of hindsight relabeling.
>
> > "how well does AMAGO prioritize or sequence goals when given multiple objectives?"
>
> When we use multi-step goals, the goal becomes an instruction or a sequence of steps, and agents receive rewards for completing goals in the assigned order. There can be an element of planning involved here where an agent could take proactive measures to prepare for later goals before it completes earlier ones. We explore this in Appendix C.5 “Planning and Long-Horizon Tasks” and Appendix C.4.2. The main benefits of multi-step goals are that 1) the user can break down a complex task into multiple steps (like how "make stone pickaxe, collect iron" can be successful when "collect iron" alone is not ), and that 2) the experience generated by attempting multi-step goals drives exploration of new goals with intermediate steps.
>
> > "Are there any experimental results that can support the claim that AMAGO does not require much tuning?"
>
> The main result here is the number of different benchmarks we apply AMAGO to in this paper with few changes in hyperparameters. We note that we do not mean AMAGO cannot benefit from tuning; hyperparameters like the learning schedule/update-to-data ratio are still important, and the use of Transformers on long sequences opens up interesting questions (Appendix D). Instead, we mean that the details of AMAGO are designed to make tuning less critical by providing a clear direction where there are many safe choices. For example, we demonstrate that AMAGO can safely learn from extra long sequences (by using all available information in environments where prior work does not) and high discount factors (by always using $\gamma \geq .999$). We avoid unintuitive hyperparameters like entropy constraints and randomize over details like the $\epsilon$-greedy schedule.
>
> > "The author mentioned replacing ReLU/GeLU with Leaky ReLU, adding LayerNorms, and modifying linear layers. It may be better to explain the necessity and rationality behind each modification through a more thorough ablation study."
>
> Our revision updates the description of these details in the method section (Page 5). These ideas are included in the POPGym ablation, where they prevent sudden collapse in performance. Our architecture decisions are motivated by observing activations in the Transformer layer before and after collapse and justified by noting that many environments require specific memory patterns that encourage instability. Following your suggestion, we have recreated an example of our early experiments that led to these decisions in a new Appendix Figure 15.

---

> > ### Comment · Reviewer_KbVB · 2023-11-22
> >
> > Thanks for the author's response. Most of my concerns have been addressed, for now I will maintain my current score and continue to pay attention to other reviews and discussions.

---

### Author Response · Authors · 2023-11-17
**Summary of Revisions**

We would like to thank all of the reviewers for their detailed comments and questions. We will respond to each individually below, but here we provide a summary of the changes made to our work. Edits to the pdf have been highlighted in blue.

### Writing

- We have updated explanations of two of our method’s technical details in Section 4 (paragraphs “Stable Long-Context Transformers in Off-Policy RL” and “Long Horizons and Multi-Gamma Learning”) which we hope make the motivation behind these ideas more clear.

- We have added an Algorithm box and expanded the discussion of multi-goal hindsight relabeling in Appendix B (as suggested by reviewer t9aE).

- Minor edits for clarity and space efficiency.

### Experiments

- Our core agent can be used with any seq2seq memory model, although our paper focuses on Transformers and the technical challenges involved in training them. We add an ablation of our agent that directly replaces the Transformer with an RNN, which helps motivate the need for Transformers in this area. This RNN ablation is used in a new figure (Figure 4) that directly evaluates the impact of Transformers in memory-intensive POPGym tasks. It has also been added to two of the meta-RL benchmarks discussed in Figure 5 (Wind, and Dark Key-To-Door in the Appendix).

- Figure 9 (the maze domain with randomized continuous action spaces) has been moved to the Appendix due to space constraints. Figure 4 (training speed profiling) has been rearranged for formatting reasons and is now Figure 6.

- Additional Appendix A.3 figure and discussion related to the Transformer architecture as requested by reviewer KbVB.

---

### Meta-Review · Area_Chair_AttJ · 2023-12-07

**Metareview:**

This paper introduces AMAGO which addresses challenges associated with generalization, long-term memory, and meta-learning by employing sequence models. There are two main contributions. Firstly, it modifies the off-policy in-context approach by incorporating sequence models like Transformers instead of recurrent policies. This modification helps overcome constraints related to memory capacity, model size, and long horizon. Secondly, AMAGO extends in-context learning to goal-conditioned problems, enabling it to handle more complex exploration tasks. Empirical evaluations on several benchmarks show its superior performance over baselines.

Overall, this is a solid paper and makes substantial contributions aforementioned to context-based meta-RL. All the questions and issues were resolved by the authors' responses. The reviewers reached a consensus for acceptance.

**Justification For Why Not Higher Score:**

The problem setting is not novel, as applying transformers to POMDPs or for in-context RL/context-based Meta RL has been extensively studied.

**Justification For Why Not Lower Score:**

This paper makes solid contributions though it is empirical.

---

### Decision · Program_Chairs · 2024-01-16

Accept (spotlight)